# Therapeutic Applications of Mesenchymal Stem Cell Loaded with Gold Nanoparticles for Regenerative Medicine

**DOI:** 10.3390/pharmaceutics15051385

**Published:** 2023-04-30

**Authors:** Wen-Yu Cheng, Meng-Yin Yang, Chun-An Yeh, Yi-Chin Yang, Kai-Bo Chang, Kai-Yuan Chen, Szu-Yuan Liu, Chien-Lun Tang, Chiung-Chyi Shen, Huey-Shan Hung

**Affiliations:** 1Department of Minimally Invasive Skull Base Neurosurgery, Neurological Institute, Taichung Veterans General Hospital, Taichung 407204, Taiwan; yangmy04@gmail.com (M.-Y.Y.); jean1007@gmail.com (Y.-C.Y.); kaiyuan.chen@vghtc.gov.tw (K.-Y.C.); syliu@vghtc.gov.tw (S.-Y.L.); chienluntang@gmail.com (C.-L.T.); shengeorge@yahoo.com (C.-C.S.); 2Department of Physical Therapy, Hung Kuang University, Taichung 433304, Taiwan; 3Institute of Biomedical Sciences, National Chung Hsing University, Taichung 402202, Taiwan; 4Department of Post-Baccalaureate Medicine, College of Medicine, National Chung Hsing University, Taichung 402202, Taiwan; 5Graduate Institute of Biomedical Science, China Medical University, Taichung 404333, Taiwan; hs0603@gmail.com (C.-A.Y.); kbwork2021@gmail.com (K.-B.C.); 6Translational Medicine Research, China Medical University Hospital, Taichung 404327, Taiwan

**Keywords:** gold nanoparticle, mesenchymal stem cell, biological performance, nanodrug delivery

## Abstract

In the present study, the various concentrations of AuNP (1.25, 2.5, 5, 10 ppm) were prepared to investigate the biocompatibility, biological performances and cell uptake efficiency via Wharton’s jelly mesenchymal stem cells and rat model. The pure AuNP, AuNP combined with Col (AuNP-Col) and FITC conjugated AuNP-Col (AuNP-Col-FITC) were characterized by Ultraviolet–visible spectroscopy (UV-Vis), Fourier-transform infrared spectroscopy (FTIR) and Dynamic Light Scattering (DLS) assays. For in vitro examinations, we explored whether the Wharton’s jelly MSCs had better viability, higher CXCR4 expression, greater migration distance and lower apoptotic-related proteins expression with AuNP 1.25 and 2.5 ppm treatments. Furthermore, we considered whether the treatments of 1.25 and 2.5 ppm AuNP could induce the CXCR4 knocked down Wharton’s jelly MSCs to express CXCR4 and reduce the expression level of apoptotic proteins. We also treated the Wharton’s jelly MSCs with AuNP-Col to investigate the intracellular uptake mechanisms. The evidence demonstrated the cells uptake AuNP-Col through clathrin-mediated endocytosis and the vacuolar-type H^+^-ATPase pathway with good stability inside the cells to avoid lysosomal degradation as well as better uptake efficiency. Additionally, the results from in vivo examinations elucidated the 2.5 ppm of AuNP attenuated foreign body responses and had better retention efficacy with tissue integrity in animal model. In conclusion, the evidence demonstrates that AuNP shows promise as a biosafe nanodrug delivery system for development of regenerative medicine coupled with Wharton’s jelly MSCs.

## 1. Introduction

Nanomedicine provides novel prospects for clinical applications to treat patients suffering from various diseases. In cancer therapy, conventional medicine may have lower specificity or may cause side effects, such as induction of inflammatory responses that lead to insufficient therapeutic treatments [1]. In regenerative medicine, the inadequate mechanical properties of artificial grafts or scaffolds may have negative biocompatible performances that reduce the interaction between wound sites and grafts/scaffolds [2]. Thus, there has been a trend in recent years to combine medical development with nanotechnology to explore possibilities for therapeutic improvement [3,4]. The surface modification of scaffolds at the nanoscale can improve the microenvironment with biomimetic properties to achieve better biocompatibility and cells/scaffolds interaction [5]. It has been verified that dopamine-patterned poly(dimethylsiloxane) via plasma treatment facilitated the differentiation capacity of mesenchymal stem cells (MSC) and provided an appropriate microenvironment for cell adhesion [6]. Additionally, drug delivery systems for cancer therapies combined with nanotechnology are increasingly attracting attention and have been investigated for decades. The literature has demonstrated that gold nanoparticles combined with Type I collagen can serve as the core system for carrying Avemar, which induces cytotoxicity in transformed SCC oral cancer cells but has better biocompatibility for normal bovine aorta endothelial cells [7]. Therefore, advanced development of clinical therapies coupled with nanotechnology has enormous potential in modern medicine. 

Inorganic nanomaterials such as graphene oxide [8], gold, silver [9] and copper [10] nanoparticles have been widely investigated in biomedicine studies. Gold nanoparticle (AuNP) has been proved to be a biocompatible nano-metallic material, which has been widely applied in various fields, including bioimaging, drug delivery core systems and surface modification techniques [11]. AuNP can be manufactured via chemical and physical processes. The AuNP obtained from chemical methods is often conducted by using tetrachloroauric acid with sodium citrate, sodium borohydride or other toxic reagents as reducing and stabilizing agents for the synthesis process [12,13]. One study reported that the nanoparticles synthesized via green approaches demonstrated lower cytotoxicity than chemical methods [14]. Therefore, the eco-friendly methods can be suggested as the promising approach to obtaining nanoparticles [15]. With respect to physical manufacture of AuNP, the process involves the physical vapor deposition (PVD) process or the physical crushing method [16]. The AuNP acquired from the PVD process possesses better biocompatibility and low toxicity because of non-chemical residues on the surface of AuNP. In our previous research, the surface of graphene oxide modified with an appropriate amount of AuNP demonstrated better anti-inflammation ability and strengthened MSC to express neuron protein markers (e.g., GFAP, nestin) [8]. Regarding the drug carrier efficiency of AuNP, it was demonstrated that AuNP could carry specific peptides to induce MSC differentiation capacity [17]. Furthermore, the AuNP-Col system involved in carrying epidermal growth factor receptor siRNA showed remarkable anti-lung cancer efficiency in the A549 cell line, with a specific cytotoxic effect [18].

Mesenchymal stem cells (MSC) can be induced by various growth factors to differentiate into neuron, bone and endothelial cells, which are considered to have multi-lineage differentiation for tissue regeneration engineering as well [19]. MSCs are readily acquired from most tissues, including umbilical cord and bone marrow, and have been proven to be hypoimmune-induced multipotent stem cells that can be used for treatment of clinical diseases [20]. In tissue engineering, the biological performance induced by biomaterials is important, particularly the relationship between tissue regeneration rate and cell proliferation/migration ability. Matrix metalloproteinases (MMPs) are highly expressed in cells and degrade extracellular matrix (ECM) during cell migration, especially MMP-2 and MMP-9 [21]. It was demonstrated that the expression of MMP-2/9 in MSC could be significantly induced by AuNP-modified collagen and fibronectin nanocomposites [22,23]. Moreover, polyurethane modified with AuNP was verified to activate the α5β3/FAK/Rho-GTPase and PI3K/Akt/eNOS pathways in endothelial cells and mesenchymal stem cell to promote proliferation and migration ability [24,25], which indicates its potential for tissue regeneration.

Cells can uptake molecules or substances that cannot directly enter the cell membrane via endocytosis [26]. The endocytosis pathways can be categorized into four types: clathrin-mediated endocytosis, caveolae-mediated endocytosis, macropinocytosis and phagocytosis. The diameter and modification properties of nanoparticles are taken up by cells via different endocytosis pathways. In the literature, it has been demonstrated that vitamin B12-modified trimethyl chitosan nanoparticles are absorbed by cells undergoing clathrin-mediated and caveolae-mediated endocytosis [27]. Additionally, iron oxide nanoparticles could be taken up by cells via clathrin-mediated endocytosis and macropinocytosis [28]. A previous study demonstrated the mechanisms whereby MSCs absorb silver nanoparticles, which have been associated with clathrin-mediated endocytosis and macropinocytosis [29]. AuNPs have recently emerged as an ideal nanocarrier for drug delivery due to their unique optical, physical, and chemical properties. These nanoparticles have a high surface area to volume ratio, making them highly reactive and amenable to functionalization with various biomolecules, including drugs and imaging agents [30]. Coupling gold nanoparticles with mesenchymal stem cell (MSC) treatments has shown great promise for enhancing drug delivery to the desired site of action [31]. Simultaneously, MSCs possess a natural tropism for inflamed or diseased tissues and can be used as vehicles to transport NPs loaded with drugs to these sites [32]. In addition, MSCs can also help protect AuNPs from clearance by the immune system, thereby enhancing their therapeutic efficacy [33]. In this study, we aimed to evaluate improvements in biological performance achieved by appropriate concentrations of AuNPs via the SDF-1/CXCR4 signaling pathway, which can influence MSC migration and proliferation. Furthermore, the uptake pathways were investigated to understand the mechanisms of AuNP absorption by MSCs, which could be valuable for future nanodrug development. 

## 2. Materials and Methods

### 2.1. Nanoparticle Preparation and Characterization 

#### 2.1.1. Preparation of AuNP and AuNP-Col 

We obtained 99.99% purity of gold nanoparticle (AuNP, 50 ppm) solution from Gold NanoTech, Inc. (GNT, Taipei, Taiwan), which was manufactured using the latest method, physical vapor deposition (PVD) processing. The gold nanoparticle solution was free from chemical compounds. The AuNP solution was filtered with a 0.22 μm filter, which was considered to be sterile. To prepare different concentrations (1.25, 2.5, 5 and 10 ppm) of AuNPs for further experiments, the AuNP solution was diluted with culture medium by the M1V1 = M2V2 mass conservation equation (M: concentration of solution, V: volume of solution).

For the preparation of AuNP-Col, each as-prepared AuNP solution (100 μL) was mixed with 100 μL of collagen (0.5 mg/mL, BD Bioscience, San Jose, CA, USA) for 30 min at room temperature (RT). Additionally, the AuNP-Col was conjugated with FITC (0.5 mg/mL, Sigma-Aldrich, St. Louis, MI, USA) at a 50:1 vol ratio for 8 h at 4 °C, which was applied for fluorescence observation.

#### 2.1.2. Ultraviolet–Visible Spectroscopy (UV-Vis) 

The pure AuNP and AuNP-Col-FITC solution were subjected to UV-Vis spectroscopy assay. The absorption spectra were obtained via the Helios Zeta spectrophotometer (ThermoFisher, Waltham, MA, USA), while the wavelength was measured from 190 to 1100 nm. The data were analyzed by Origin Pro 8 software (Originlab Corporation, Northampton, MA, USA).

#### 2.1.3. Dynamic Light Scattering (DLS) 

The size distribution intensity and hydrodynamic diameter of pure AuNP, AuNP-Col and AuNP-Col-FITC were analyzed by dynamic light scattering (DLS) assay. The analysis was conducted using a Malvern Zetasizer Nano ZS instrument (Malvern Panalytical Ltd., Malvern, UK) and manipulating a 633 nm He–Ne laser with a 90° scattering angle. To execute the experiment, 1 mL of each colloidal sample was added into a 1 cm optical path cuvette at 25 °C for examinations.

#### 2.1.4. Fourier-Transform Infrared Spectroscopy (FTIR) 

The pure AuNP, pure col and AuNP-Col nanomaterials were subjected to Fourier Transformed Infrared (FTIR) spectrometry (Shimadzu FTIR Model IRPrestige-21, Shimadzu Corporation, Kyoto, Japan). The functional groups on the surface of each sample were detected, and the spectrum was measured at the absorption range of 400–4000 cm^−1^. One percent (*w/w*) of each sample was cautiously mixed with 100 mg of potassium bromide (KBr) powder (Sigma, St. Louis, MO, USA) for further pressing into a sheer slice. A total of 32 scans were conducted for each sample to improve the signal-to-noise percentage.

### 2.2. Cell Culture 

The human umbilical cord Wharton’s jelly mesenchymal stem cells (MSCs) were applied for the present research, which were also selected in our previous articles [9,34]. The cells were stored after being cultured in H-DMEM medium (Invitrogen, Waltham, MA, USA) containing 10% fetal bovine serum (FBS) and 1% (*v/v*) antibiotics (100 U/mL penicillin/streptomycin). The cells at the 8th passage used for various experiments were cautiously cultured in a 37 °C incubator with 5% CO_2_ humidified atmosphere. 

To identify the phenotypes of Wharton’s jelly MSCs, the cells were detached by 2 mM EDTA with PBS. The MSCs were washed by PBS containing 2% bovine serum albumin (BSA) and 0.1% sodium azide (Sigma, USA). Next, the cells were cultured with various specific antibodies including CD14-FITC-A, CD34-FITC-A, CD45-FITC-A, CD29-FITC-A, CD44-PE-A and CD73-PE-A (fluorescein isothiocyanate represented FITC and phycoerythrin denoted as PE). The FITC/PE conjugated IgG1 were used for isotype controls (BD Pharmingen, San Diego, CA, USA). A flow cytometer was applied for MSC phenotype detection. 

### 2.3. CXCR4 siRNA Transfection in MSCs 

The CXCR4 siRNA (12.5 nM, Santa Cruz Bicycles, Santa Cruz, CA, USA) were transfected into Wharton’s jelly MSCs through lipofectamine to knock down the CXCR4 expression. To prepare the 12.5 nM CXCR4 siRNA, the CXCR4 siRNA was mixed with lipofectamine in a 2:1 vol ratio for 30 min at RT, and then diluted with Opti-MEM medium for 1:1000 dilution. After removing the previous medium, the transfection solution was added into the culture plate for 12–16 hours’ incubation to knock down the CXCR4 expression. Next, the MSCs were assigned into the following treatments: Group 1: No AuNP (0 ppm), 50 ng of SDF-1α treatment with no AuNP and CXCR4 siRNA + 50 ng of SDF-1α treatment with no AuNP; Group 2: AuNP 1.25 ppm, 50 ng of SDF-1α treatment with 1.25 ppm AuNP and CXCR4 siRNA + 50 ng of SDF-1α treatment with 1.25 ppm AuNP; Group 3: AuNP 2.5 ppm, 50 ng of SDF-1α treatment with 2.5 ppm AuNP and CXCR4 siRNA + 50 ng of SDF-1α treatment with 2.5 ppm AuNP. The MSCs cultured with various treatments were collected for the investigations of cell viability, metalloproteinase expression, migration ability, CXCR4 expression and apoptosis. All of the experiments were executed in triplicate.

### 2.4. Measurement of Cell Viability 

To investigate the cell viability of Wharton’s jelly MSCs that were treated with various treatments, as described in Section 2.3, we conducted MTT [3-(4,5-dimethylthiazol-2-yl)-2,5-diphenyltetrazolium bromide] assay based on the manufacturer’s instructions (Sigma-Aldrich, USA). MSCs at a cell density of 1 × 10^4^ per well were cultured in 96-well culture plates with various treatments for 48 h. Next, 100 μL of MTT solution (0.5 mg/mL) was added into each well and incubated at 37 °C, 5% CO_2_ for 2 h. Next, 100 μL of Dimethyl sulfoxide (DMSO) solution was added and incubated for 10 min. The absorbance at 570 nm was detected by SpectraMax M2 ELISA reader (Molecular Devices, San Jose, CA, USA). The AuNP 0 ppm group served as the Control group.

### 2.5. Investigation of Biological Performance within MSC 

#### 2.5.1. CXCR4 Expression 

The Wharton’s jelly MSCs (1 × 10^4^ cells/well) were seeded in the 24-well culture plates containing 15 mm glass coverslips and cultured at 37 °C, 5% CO_2_ atmosphere. After incubating overnight for cell attachment, we treated MSCs with various treatments, as described in Section 2.3, and incubated them for 48 h. Next, the cells were washed thrice by PBS solution, fixed with 4% paraformaldehyde (PFA) at 4 °C for 30 min, and washed three times by PBS solution. Afterwards, 0.5% Triton X-100 was added for 10 min incubation at RT, and then 5% FBS was added for 30 min blocking at 4 °C. To stain the cell skeleton, the phalloidin solution was first diluted with PBS at 1:400 vol ratio, and then applied for incubating with MSC in a dark room for 1 h at RT. For determination of CXCR4 expression, the CXCR4 primary antibody (1:500 dilution) was added for 8 h incubation at 4 °C, and then fluorescein isothiocyanate-conjugated goat anti-mouse secondary IgG antibodies (1:200 diluted by PBS) was added for 1 h incubation at RT. Ultimately, DAPI (4,6-diamidion-2-phenylindole) solution (1:5000 diluted by PBS) was added for 10 min to locate cell nuclei. The cells were washed thrice by PBS solution prior to each step. The green fluorescence of CXCR4 expression in MSC was observed by Zeiss Axio Imager A1 fluorescence microscope (Pentland Cre., Vaughan, ON, Canada), and the fluorescence intensity was quantified by Image J 5.0 software (National Institutes of Health, Bethesda, MD, USA). The fluorescein-positive cells were also detected by a flow cytometer, and the data were analyzed by fluorescence-activated cell sorting (FACS) software (BD Biosciences, San Jose, CA, USA).

#### 2.5.2. Gelatin Zymography Analysis 

The 2 × 10^5^ cells/well of Wharton’s jelly MSCs were cultured in 6-well culture plates and incubated overnight (37 °C with 5% CO_2_ atmosphere) to achieve cell attachment. Next, the cells were treated with various treatments, as described in Section 2.3. After incubation for 48 h, the samples containing cells were centrifuged and the supernatants were collected to determine MMP-2 and MMP-9 activities, following the protocols described in the previous literature [23]. In brief, the samples were separated by 10% zymogram protein gel containing 0.1% gelatin, and then the gelatin-degrading proteolytic activities of MMP-2 and MMP-9 on the gel could be clearly observed within the dark blue background stained by Coomassie Blue solution. After the gels were destained by 10% acetic acid and 40% ethanol solution, the gels were scanned, and the MMPs expression was analyzed by Gel-Pro Analyzer 4.0 software (Media Cybernetics, Rockville, MD, USA).

#### 2.5.3. Cell Migration Ability 

The Wharton’s jelly MSCs (1 × 10^4^ cells/well) were seeded in 96-well culture plates and cultured at 37 °C, 5% CO_2_ atmosphere. After overnight incubation for cell attachment, MSCs were treated with various treatments, as described in Section 2.3, then incubated for 48 h. The migration ability of MSC was determined by Oris Cell Migration Assay reagent kit (Platypus Technologies, Madison, WI, USA) [23]. Next, 2 µM of Calcein AM solution was added to stain the MSC for 30 min incubation. The images of migration ability were observed and captured by a Zeiss Axio Imager A1 fluorescence microscope. The migration distance of each treatment was quantified based on the fluorescence intensity in the detection area by Image J 5.0 software.

### 2.6. Measurement of Apoptotic-Related Protein Expression 

The 2 × 10^5^ cells of Wharton’s jelly MSCs were seeded in 10 cm^2^ culture plate and incubated overnight at 37 °C with 5% CO_2_ atmosphere for cell attachment. Next, the cells were treated with various treatments, as described in Section 2.3, followed by incubation for 48 h. After incubation, the MSCs were collected by 0.05% trypsin-EDTA solution, washed two times by PBS solution, and pH 7.5 RIPA buffer (Tris: 5 mL, NaCl: 3 mL, NP-40: 1 mL, 10% SDS: 1 mL, sodium deoxycholate: 0.5 g) was added for 1 h incubation to lyse the MSCs. Next, the supernatant with proteins was acquired by 20 min centrifugation (4 °C, 13,000 rpm). Afterwards, the protein concentration was determined using a BCA Protein Assay Reagent Kit based on the manufacturer’s instructions (Bio-Rad Laboratories Inc., Hercules, CA, USA). Each sample containing proteins was electrophoresed in an SDS-PAGE gel to separate the proteins (120 voltage, 120 min). To transfer the proteins onto PVDF membranes (Immobilon P; EMD Millipore), the transfer buffer solution (Tris base: 3 g, 100% methanol: 150 mL, glycine: 14.4 g, ddH_2_O: 1000 mL) was added into a Bio-Rad Mini Protean System device, and the process was conducted on ice for 1 h (400 mA). After the protein transfer process, the PVDF membranes were blocked in TBST solution (Tris-HCl: 60.57 g, NaCl: 85 g, Tween-20: 5 mL, ddH_2_O: 1000 mL) with 5% milk powder and subjected to shaking for 1 h. The membranes were probed with primary antibodies at 4 °C for overnight incubation (1:1000 dilution of Caspase-3, p21, Bax, Bcl-2, Cyclin-D1; 1:2000 dilution of β-actin (Santa Cruz Bicycles, USA)). Next, the membranes were washed three times by TBST solution, and then incubated with HRP-conjugated goat anti-rabbit or anti-mice IgG (1:2000 dilution, Zhongshan Goldenbridge Biotechnology, Beijing, China) for 1 h at RT. Finally, an ECL kit (PerkinElmer, Waltham, MA, USA) was applied for observation of each protein band under X-ray. The expression of each of the protein samples was analyzed with a Gel-Pro Analyzer 4.0 (Media Cybernetics, USA). Each protein band density was normalized to β-actin.

### 2.7. Exploration of Cellular Uptake Mechanisms 

#### 2.7.1. Intracellular Uptake Investigation 

The Wharton’s jelly MSCs (1 × 10^4^ cells/well) were seeded on 24-well culture plates with 15 mm glass coverslips for overnight incubation. After cell attachment, the MSCs were incubated with AuNP-Col-FITC (Control: no treatment; AuNP: 1.25 ppm, 2.5 ppm, 5 ppm, 10 ppm) for 30 min, 2 h, and 24 h at 37 °C with 5% CO_2_ atmosphere. After the incubation, the cells were washed with PBS solution three times, fixed with 4% paraformaldehyde (PFA) for 30 min, washed thrice with PBS, and 0.5% Triton X-100 (Sigma-Aldrich, USA) was added for permeability for 10 min. The FBS solution was added into each sample for blocking (4 °C, 30 min). Next, the cell skeleton (F-actin) was stained by 1:400 dilution of Rhodamine phalloidin (Sigma-Aldrich) for 1 h in the dark, while the cell nuclei were stained by 4,6-diamidino-2-phenylindole (DAPI, 1 mg/mL, Invitrogen, USA) solution for 10 min and washed thrice by PBS solution. Finally, the fluorescence was observed using a Zeiss Axio Imager A1 fluorescence microscope and analyzed by Image J 5.0 software. The fluorescein-positive cells were also detected with a flow cytometer to analyze cell uptake efficiency by fluorescence-activated cell sorting (FACS) software (BD Biosciences, USA). The experiments were executed in triplicate.

#### 2.7.2. LysoTracker Analysis 

To investigate the endosomal/lysosomal escape for nanoparticle delivery, LysoTracker assay was applied for intracellular delivery. Wharton’s jelly MSCs (1 × 10^4^ cells/well) were cultured in 24-well culture plates with 15 mm glass coverslips for overnight incubation to reach cell attachment. Next, the MSCs were treated with different treatments: Control group (no treatment) and AuNP-Col-FITC (2.5 ppm of AuNP, green fluorescence) for 30 min, 2 h and 24 h. Next, the LysoTracker red fluorescent probes (50 nM, Life Technologies, Carlsbad, CA, USA) were added for 1 h incubation and washed thrice with PBS solution. The cells were fixed with 4% paraformaldehyde (PFA) for 30 min at 4 °C, washed thrice with PBS, and 0.5% Triton X-100 was added for permeability for 10 min. The cell nuclei were located by DAPI (1 mg/mL) solution for 10 min in the dark and washed thrice by PBS. Finally, the fluorescence was observed using a Zeiss Axio Imager A1 fluorescence microscope and analyzed by Image J 5.0 software. The experiments were executed in triplicate.

#### 2.7.3. Assessments of Endocytic Pathways 

The endocytic mechanisms of Wharton’s jelly MSCs were investigated by culturing with various specific inhibitors. MSCs (1 × 10^4^ cells per well) were seeded in 24-well culture plates with 15 mm glass coverslips. After overnight incubation for cell attachment, the MSCs were first treated with four specific inhibitors for 1 h: Chlorpromazine (CPZ, 2 μM), Cytochalasin D (CCD, 5 μM), Bafilomycin A (Baf, 100 nM) and Methyl-β-cyclodextrin (MCD, 2.5 mM) (Sigma-Aldrich, USA). The inhibitors were prepared with 10% DMSO solution prior to dilution with cell culture medium. The optimal concentration of each inhibitor treating MSCs was determined by MTT assay. Next, the MSCs were pre-treated with a specific concentration of each inhibitor described above for 1 h. Afterwards, the cells were treated with AuNP-Col-FITC (2.5 ppm of AuNP, green fluorescence) for 30 min, 2 h and 24 h. The fluorescence was observed using a Zeiss Axio Imager A1 fluorescence microscope and analyzed by Image J 5.0 software. The fluorescein-positive cells were detected by a flow cytometer and analyzed by FACS software. The experiments were executed in triplicate.

### 2.8. Mice Model 

#### 2.8.1. In Vivo Subcutaneous Implantation 

The female Sprague-Dawley (SD) rats used in this research (2–3 months, 350 g) were purchased from the National Laboratory Animal Center (Taipei, Taiwan). The experiments were executed according to National Institute of Health guidelines with the approval of China Medical University’s Animal Care and Use Committee (CMUIACUC-103-90-N). In the experiments, AuNP 2.5 ppm solution was first coated onto 15 mm glass coverslips. The coverslips with AuNP coating were implanted onto the rat subcutaneous tissues. The mice were given local anesthesia, and then the coatings were implanted into the dorsal skin of rats (10 mm^2^ area). After one month of implantation, the rats (*n* = 5) were sacrificed. The tissues containing implanted materials were acquired for various histological examinations. The fibrous capsule thickness from six sites induced by the as-prepared materials was explored by hematoxylin and eosin (H&E) staining assay, and the thickness of fibrotic tissues were quantified using Image J 5.0 software. The collagen deposition in the tissues was evaluated by Masson’s trichrome staining assay (Sigma-Aldrich, USA), and the deposition amount was also quantified by Image J software. To investigate the endothelialization induced by as-prepared materials in tissues, monoclonal mouse anti-CD31 antibodies were assessed in the experiment. To detect apoptotic cells in tissues, TUNEL assay was applied in the investigation. We purchased an In Situ Cell Death Detection Kit, AP (Roche Diagnostics, Indianapolis, IN, USA) to detect apoptotic cells according to the manufacturer’s protocol. The fluorescence intensity was observed using an Olympus ix71 fluorescence microscope. The nuclear DNA was targeted with DAPI solution. The data from each experiment are expressed as mean ± SD.

#### 2.8.2. In Vivo Tissue Integrity and Particle Distribution 

Approval for the use of the female Sprague-Dawley (SD) rats in our experiments was obtained from China Medical University’s Animal Care and Use Committee (CMUIACUC-103-90-N). AuNPs were first conjugated with FITC to further direct injection into the retroorbital sinus of the mice. The mice were sacrificed at 24 and 48 h. Next, the organs/tissues, including brain, heart, liver, spleen, lung and kidney, were cautiously acquired and fixed by 4% paraformaldehyde (PFA). The collected tissues were dehydrated and embedded in paraffin. To evaluate the histological examination of tissue integrity, the tissues were cautiously cryosectioned into slices measuring 4 μm thick for hematoxylin and eosin (H&E) staining (Sigma, USA). The particle distribution of AuNP was observed by fluorescence microscopy based on the green fluorescence.

### 2.9. Statistical Analysis 

To avoid uncertainty, the experiments were conducted three times independently, and the results are expressed as mean ± standard deviation (SD). SPSS Statistics 17.0 software (IBM Inc, Armonk, NY, USA) was used for the exploration of statistical differences. A *p* value less than 0.05 was considered statistically significant.

## 3. Results

### 3.1. Characterization of AuNP-Derived Nanomaterials

In the present research, we prepared various nanomaterials which were presented as pure AuNP and AuNP-Col. The AuNP-Col were conjugated with FITC for the investigation of cell metabolisms. Figure 1A shows the UV-Vis spectrum, with specific absorbance peak of AuNP at the wavenumber of 520 nm in each sample. Figure 1B shows the FTIR spectra of pure Col, pure AuNP and AuNP-Col. According to the previous literature published in 1989, the IR measurement from liquid water demonstrated the bands at around 3400 cm^−1^ (symmetric and antisymmetric stretch of O-H bonding) and 1648 cm^−1^ (H–O–H bending) [35]. Another study published in 2004 showed the FTIR results of AuNP produced in deionized water, which reported that the stretch bonding of carbonato complexes (Au–OCO_2_− and Au–OCO_2_H) on AuNP at 1000, 1334 and 1595 cm^−1^ [36] due to atmospheric CO_2_ is slightly soluble in water and can form HCO_3_^−^ and H^+^ during the preparation of samples. Moreover, DLS assay was conducted for the examination of size distribution intensity (Figure 1C) and nanoparticle diameter (Figure 1D). The quantitative results demonstrated that the average diameter of AuNP was 25.36 ± 8.2 nm, AuNP-Col was 90.95 ± 23.3 nm and AuNP-Col-FITC was 255.96 ± 25.1 nm. The TEM images for Au and AuNP-Col could be referenced in our published study, while the polydispersity index was also included in [37].

### 3.2. CXCR4 Expression Induced by AuNP in Wharton’s Jelly MSC

The phenotypes of Wharton’s jelly MSCs used in this study were identified by CD14, CD29, CD34, CD44, CD45 and CD73 surface markers. Appendix A displays the flow cytometric results of each specific marker expression. The expression of CD14, CD34 and CD45 endothelial markers was analyzed as 0.43%, 0.81% and 0.335% (negative markers), respectively. The cells expressed CD29, CD44 and CD73 as 97.7%, 93.6% and CD73% (positive markers), respectively, which demonstrated the expression of MSC surface markers.

To investigate the CXCR4 expressed in Wharton’s jelly MSCs through the induction of AuNP, we first knocked down the CXCR4 expression with CXCR4 siRNA (CXCR4 si) transfection. The process of CXCR4 si transfection was executed for 12–16 h, and the expression intensity was observed with a fluorescence microscope (Appendix A). The expression of CXCR4 was quantified as Control (1-fold) and CXCR4 si (~0.36-fold, ** *p* < 0.01) groups, demonstrating that MSCs expressed lower CXCR4 after the transfection (Appendix A).

The Wharton’s jelly MSCs were subjected to AuNP 0 ppm, 1.25 ppm and 2.5 ppm treatments to measure the cell viability. Figure 2 displays the cell viability of MSC with various treatments for 48 h, which was quantified as: AuNP 0 ppm treatment (No AuNP: 1.18, SDF-1α: 1.26, CXCR4 si + SDF-1α: 0.63), AuNP 1.25 ppm treatment (pure AuNP: 1.34, SDF-1α: 1.49, CXCR4 si + SDF-1α: 1.13), and AuNP 2.5 ppm treatment (pure AuNP: 1.39, SDF-1α: 1.41, CXCR4 si + SDF-1α: 0.99). According to the results, the MSC viability could be strengthened by pure AuNP 1.25 ppm and AuNP 2.5 ppm. After treatment with 50 ng of SDF-1α, the MSC viability was greater in both AuNP 1.25 ppm and AuNP 2.5 ppm groups. After knockdown of CXCR4 expression, the viability was still lower after treatment with 50 ng of SDF-1α in the AuNP 0 ppm group. However, the MSC proliferation with CXCR4 siRNA + SDF-1α treatment could be significantly facilitated by the addition of AuNP 1.25 and 2.5 ppm. It was demonstrated that AuNP at concentrations of 1.25 and 2.5 ppm were appropriate solutions for MSC proliferation.

To explore the relationship between SDF-1α/CXCR4 and AuNP, we further investigated the expression of CXCR4 in Wharton’s jelly MSCs with the induction of AuNP. The MSCs were treated with AuNP 0 ppm, 1.25 ppm and 2.5 ppm solution for 48 h, and the CXCR4 green fluorescence intensity was captured and analyzed by IF and FACS methods. Figure 3A displays the CXCR4 expression in MSC enhanced by AuNP, and the FACS histograms are demonstrated in Appendix A. The results, quantified based on the CXCR4 fluorescence intensity, are shown in Figure 3B: AuNP 0 ppm treatment (No AuNP: ~1-fold, SDF-1α: ~1.29-fold, CXCR4 si + SDF-1α: ~0.60-fold), AuNP 1.25 ppm treatment (pure AuNP: ~1.59-fold, SDF-1α: ~2.10-fold, CXCR4 si + SDF-1α: ~1.03-fold), and AuNP 2.5 ppm treatment (pure AuNP: ~1.72-fold, SDF-1α: ~2.12-fold, CXCR4 si + SDF-1α: ~0.91-fold). Furthermore, the CXCR4 fluorescein-positive cells were detected with flow cytometry, and results are displayed in Figure 3C: AuNP 0 ppm treatment (No AuNP: ~1-fold, SDF-1α: ~1.38-fold, CXCR4 si + SDF-1α: ~0.32-fold), AuNP 1.25 ppm treatment (pure AuNP: ~1.27-fold, SDF-1α: ~1.45-fold, CXCR4 si + SDF-1α: ~0.69-fold), and AuNP 2.5 ppm treatment (pure AuNP: ~1.19-fold, SDF-1α: ~1.43-fold, CXCR4 si + SDF-1α: ~0.47-fold). According to the quantification, although the CXCR4 expression in the MSCs was knocked down, treatment with AuNP 1.25 and 2.5 ppm could significantly enhance its expression.

### 3.3. Biological Performance of Wharton’s Jelly MSCs with AuNP Treatment

The MMPs expression, migration ability and apoptosis-related protein expression within Wharton’s jelly MSCs were investigated. The SDF-1α/CXCR4 pathway was associated with biological performances, such as cell migration. The zymogram of MMPs expression in MSCs after treatment with AuNP for 48 h is shown in Figure 4A, while the quantifications of MMP-9 and MMP-2 are shown in Figure 4B,C, respectively. The results of MMP-9 expression are as follows: AuNP 0 ppm treatment (No AuNP: ~1-fold, SDF-1α: ~1.33-fold, CXCR4 si + SDF-1α: ~0.76-fold), AuNP 1.25 ppm treatment (pure AuNP: ~1.47-fold, SDF-1α: ~1.66-fold, CXCR4 si + SDF-1α: ~1.32-fold) and AuNP 2.5 ppm treatment (pure AuNP: ~1.41-fold, SDF-1α: ~1.35-fold, CXCR4 si + SDF-1α: ~0.82-fold). The results of expression intensity of MMP-2 are as follows: AuNP 0 ppm treatment (No AuNP: ~1-fold, SDF-1α: ~1.06-fold, CXCR4 si + SDF-1α: ~0.94-fold), AuNP 1.25 ppm treatment (pure AuNP: ~1.12-fold, SDF-1α: ~1.18-fold, CXCR4 si + SDF-1α: ~1.09-fold) and AuNP 2.5 ppm treatment (pure AuNP: ~1.11-fold, SDF-1α: ~1.10-fold, CXCR4 si + SDF-1α: ~0.96-fold).

We further executed an Oris Cell Migration Assay to evaluate the efficiency of AuNP treatment. It was demonstrated that the Wharton’s jelly MSCs migration distance moving into the boundary area was longer after the AuNP 1.25 and 2.5 ppm treatment (Figure 5A). Thus, the statistical results shown in Figure 5B are as follows: AuNP 0 ppm treatment (No AuNP: 21.9 μm, SDF-1α: 25.9 μm, CXCR4 si + SDF-1α: 20.4 μm), AuNP 1.25 ppm treatment (pure AuNP: 25.6 μm, SDF-1α: 29.1 μm, CXCR4 si + SDF-1α: 25.1 μm) and AuNP 2.5 ppm treatment (pure AuNP: 23.8 μm, SDF-1α: 26.6 μm, CXCR4 si + SDF-1α: 22.4 μm). According to the evidence of MMP expression and migration ability, we confirmed that AuNP 1.25 and 2.5 ppm could significantly induce MMPs expression and further facilitate MSC migration efficiency after silencing CXCR4 expression, which was associated with the SDF-1α/CXCR4 pathway.

Apoptosis induced by biomaterials is an important issue. An appropriate biomaterial should induce lower cell apoptosis and promote cell proliferation. In this study, the expression of apoptotic-related proteins including Cyclin D1, p21, Bcl-2, Bax and Active-Caspase-3 were measured at 48 h by Western blotting assay, and the immune blots are displayed in Figure 6A. The quantitative results of each protein expression is shown in Figure 6B–F and have been organized in Table 1, Table 2, Table 3, Table 4 and Table 5, which show that MSCs had significantly lower expression of apoptotic proteins (p21, Bax, Act-caspase-3) expression after treatment with 1.25 ppm and 2.5 ppm AuNP to prevent apoptosis in Wharton’s jelly MSCs.

### 3.4. Intracellular Metabolic Mechanisms of Wharton’s Jelly MSCs with AuNP

#### 3.4.1. Cell Uptake Efficiency

To investigate AuNP uptake efficiency in Wharton’s jelly MSCs, the various concentrations of AuNP (1.25, 2.5, 5, 10 ppm) were combined with Type I collagen (Col), which is denoted as AuNP-Col. Furthermore, the AuNP-Col were conjugated with FITC for treatments with MSC followed by investigations with fluorescence microscopy and flow cytometry. The green FITC fluorescence images and FACS diagrams of AuNP-Col are shown in Figure 7A and Appendix A, while the quantitative analysis based on IF and FACS methods are shown in Figure 7B,C. The results from the IF method are as follows: 30 min (Control (0.00-fold), AuNP 1.25 ppm (~1.00-fold), AuNP 2.5 ppm (~1.57-fold), AuNP 5 ppm (~2.04-fold) and AuNP 10 ppm (~4.15-fold)); 2 h (Control (0.00-fold), AuNP 1.25 ppm (~4.14-fold), AuNP 2.5 ppm (~6.29-fold), AuNP 5 ppm (~6.57-fold) and AuNP 10 ppm (~7.34-fold)); 24 h (Control (0.00-fold), AuNP 1.25 ppm (~7.22-fold), AuNP 2.5 ppm (~8.47-fold), AuNP 5 ppm (~10.44-fold) and AuNP 10 ppm (~11.22-fold)). Additionally, the results of FACS analysis are as follows: 30 min (Control (1.00-fold), AuNP 1.25 ppm (~1.01-fold), AuNP 2.5 ppm (~1.03-fold), AuNP 5 ppm (~1.02-fold) and AuNP 10 ppm (~1.12-fold)); 2 h (Control (1.00-fold), AuNP 1.25 ppm (~1.09-fold), AuNP 2.5 ppm (~1.08-fold), AuNP 5 ppm (~1.15-fold) and AuNP 10 ppm (~1.23-fold)); 24 h (Control (1.00-fold), AuNP 1.25 ppm (~1.23-fold), AuNP 2.5 ppm (~1.34-fold), AuNP 5 ppm (~1.50-fold) and AuNP 10 ppm (~1.57-fold)). The above evidence indicates that the AuNP-Col uptake efficiency of Wharton’s jelly MSCs was significantly higher compared with the control.

#### 3.4.2. Endocytic Mechanisms of Wharton’s Jelly MSC Uptake AuNP

Four specific endocytosis inhibitors were applied for the investigation of Wharton’s jelly MSC uptake AuNP, including Chlorpromazine (CPZ, 2 μM), Methyl-β-cyclodextrin (MCD, 2.5 mM), Cytochalasin D (CCD, 5 μM), and Bafilomycin A (BAF, 100 nM). The concentration of each inhibitor mentioned above was determined by MTT cytotoxicity assay, which is shown in Appendix A. The MSCs were pre-treated with each specific inhibitor and further cultured with AuNP-Col 2.5 ppm. The MSCs’ uptake of AuNP-Col was evaluated through both IF and FACS methods. The fluorescence images and FACS histograms are depicted in Figure 8A and Appendix A. The results based on IF method (Figure 8B) are as follows: 30 min (Control (1.00-fold), CPZ (~0.57-fold), MCD (~0.99-fold), CCD (~1.01-fold) and BAF (~0.59-fold)); 2 h (Control (1.97-fold), CPZ (~1.05-fold), MCD (~1.79-fold), CCD (~1.65-fold) and BAF (~0.71-fold)); 24 h (Control (2.89-fold), CPZ (~1.29-fold), MCD (~2.23-fold), CCD (~2.02-fold) and BAF (~1.32-fold)). Additionally, Figure 8C display the FACS analysis results, which are as follows: 30 min (Control (1.00-fold), CPZ (~0.94-fold), MCD (~0.98-fold), CCD (~1.01-fold) and BAF (~0.76-fold)); 2 h (Control (1.00-fold), CPZ (~0.90-fold), MCD (~0.96-fold), CCD (~0.94-fold) and BAF (~0.85-fold)); 24 h (Control (1.00-fold), CPZ (~0.88-fold), MCD (~1.01-fold), CCD (~0.95-fold) and BAF (~0.70-fold)). The above quantifications indicate that Wharton’s jelly MSCs uptake AuNP-Col could be suppressed by CPZ and BAF inhibitors, which demonstrates involvement of clathrin-mediated endocytosis and the vacuolar-type H^+^-ATPase pathway.

#### 3.4.3. Intracellular Transportation Exploration

The stability of AuNP-Col 2.5 ppm in Wharton’s jelly MSCs was investigated by LysoTracker assay. The fluorescence images were captured at 30 min, 2 h and 24 h, as shown in Figure 9A. The semi-quantitative results are shown in Figure 9B. The results demonstrated the amount of AuNP-Col was significantly higher at 2 h and 24 h (Control: 0-fold, 30 min: ~1.00-fold, 2 h: ~1.89-fold, 24 h: ~3.69-fold), which indicates that the AuNP-Col could escape the digestion of lysosomes and remain stable in Wharton’s jelly MSCs.

### 3.5. Animal Models

#### 3.5.1. Foreign Body Responses of AuNP through In Vivo Experiments

AuNP 2.5 ppm was used in animal models to examine the induction of foreign body responses. Figure 10A shows the images of capsule formation by H&E staining, and Figure 10E shows that the capsule thickness was significantly lower in AuNP 2.5 ppm treatment (~0.68-fold, ** *p* < 0.01). The collagen deposition was further investigated, as shown in Figure 10B,F, while the deposition seen in the AuNP 2.5 ppm treatment group (~0.52-fold, ** *p* < 0.01) was remarkably lower than that of the control group. Moreover, the macrophage polarization (M1, CD86, red color; M2, CD163, green color) was evaluated through immunohistochemistry staining assay (Figure 10C,D). The fluorescence intensity was quantified and the results, displayed in Figure 10G,H, indicate a significantly lower CD86 expression (~0.52-fold, ** *p* < 0.01) but higher CD163 expression (~1.46-fold, ** *p* < 0.01). Furthermore, we conducted a TUNEL assay to investigate apoptosis cells after treatment with AuNP. The fluorescence images are displayed in Figure 11, and the statistical analysis (data not shown) demonstrated no significant difference compared to the control. It was demonstrated that AuNP 2.5 ppm did not trigger serious foreign body responses, which indicates its potential for clinical applications.

#### 3.5.2. In Vivo Biodistribution of Gold Nanoparticles

For the evaluation of tissue integrity and particle distribution, AuNP was first conjugated with FITC, and was administered into mice via retro-orbital sinus injection for in vivo assessments. The tissues/organs, including brain, heart, liver, spleen, lung, and kidney, were acquired after 12- and 24-hours treatment. The tissue morphology measured through H&E staining assay at both time points demonstrated that the AuNP treatment did not cause serious damage (Figure 12A,C). Furthermore, the fluorescence images of biodistribution indicated that the AuNP-FITC could be observed in each tissue at both time points (Figure 12B,D). According to the above results, AuNP was found to be safe and showed retention efficacy in the treatment of mice, which suggests it may have potential in the development of nanodrugs. The AuNP at concentrations of 1.25 and 2.5 ppm could strengthen the Wharton’s jelly MSC biological performance via endocytic pathways of AuNP-Col uptake is indicate in the Figure 13.

## 4. Discussion

Gold nanoparticles (AuNP) have been applied in a variety of biomedical fields due to their superior biocompatibility and ease of modification by other bioactive molecules [38]. MSCs are a cell type with potential for tissue engineering and therapeutic approaches. This study aimed to evaluate the interactions between AuNP and mesenchymal stem cells, including various biological performances and endocytosis mechanisms, with a view toward determining whether a combination of AuNP and MSCs could have possible applications in future therapies. CXCR4 expression was knocked down by the transfection of CXCR4 siRNA into MSCs, and AuNP treatments were conducted in the investigations. The SDF-1α/CXCR4 axis has been verified to be associated with cell motility in tissue [39]; moreover, SDF-1α was proven to combine with CXCR4 and CXCR7 [39]. We further treated MSCs with 1.25 ppm and 2.5 ppm of AuNP, and the cell viability of MSC was indeed significantly enhanced (Figure 2). The effect of AuNP on MSC had better proliferation and was lower for those containing either a lesser (1.25 and 2.5 ppm) amount of AuNP, suggesting that surface morphology may have more relevant homogenous distribution, which could account for better cellular adhesion and migration effect [40]. However, the AuNP contribution to overloading may be a result of aggregation. The effect at higher AuNP (5 and 10 ppm) was not evident, possibly due to the aggregation of the nanoparticles. If the dispersion of AuNP could be improved, the effect could be even more pronounced at higher AuNP concentrations. CXCR4 expression was investigated with AuNP at concentrations of 1.25 and 2.5 ppm, which could enhance the CXCR4 knockdown MSCs to highly express CXCR4 (Figure 3).

Previous research concluded that AuNPs could stimulate the expression of MMPs and trigger cell migration ability [22]. The cells secreted MMPs that degraded ECMs so they would migrate to other sites. The results from Figure 4 and Figure 5 show that CXCR4 knockdown Wharton’s jelly MSCs exhibited lower expression of MMPs, which was related to the decreased migration distance. However, after the treatment with 1.25 ppm and 2.5 ppm of AuNPs, the MMP-2/9 expression was remarkably induced, and the migration distance was longer as well. The above results indicate AuNP could positively activate the SDF-1α/CXCR4 axis to strengthen the biological performance of MSCs. The investigation of apoptotic-related protein expression was conducted. The CXCR4 knockdown Wharton’s jelly MSCs expressed lower cell cycle protein Cyclin D1 [41], and p21 protein was highly expressed (cyclin-dependent kinase inhibitor 1 [42]). Additionally, the apoptotic proteins Bax and Active-Caspase-3 were found to be expressed in CXCR4 knockdown MSCs. After treatment with 1.25 ppm and 2.5 ppm of AuNP, the CXCR4 knockdown MSCs expressed lower p21, Bax, and Act-caspase-3 proteins. In contrast, Cyclin D1 and apoptosis-inhibited protein Bcl-2 was triggered and significantly expressed. The evidence indicates that the SDF-1α/CXCR4 signaling pathway activated by 1.25 and 2.5 ppm of AuNP could promote entry of CXCR4 knockdown MSCs into the cell cycle and prevent apoptosis of the cells from.

Various endocytosis pathways are utilized in the cell to take up nanoparticles, a process that is largely affected by diameter and surface modification properties [43]. The AuNP-Col-FITC uptake efficiency was verified to be significantly higher at concentrations of 1.25 ppm and 2.5 ppm AuNP (Figure 7). We further investigated the endocytic mechanisms of MSC using four specific inhibitors: Chlorpromazine (CPZ), Methyl-β-cyclodextrin (MCD, Cytochalasin D (CCD), and Bafilomycin A (BAF). CPZ was verified to suppress clathrin-mediated endocytosis via anchoring clathrin and adaptor protein 2 (AP2) complex to endosomes [44]. MCD has been shown to function as an inhibitor to block caveolin-mediated endocytosis [45]. CCD could cause actin depolymerization to influence macropinocytosis and phagocytosis [46,47]. BAF was revealed to suppress clathrin-mediated endocytosis and caveolin-mediated endocytosis by blocking vacuolar-type H^+^-ATPase pathway, which is the down regulation of endocytosis [48]. According to our results in Figure 8, the uptake of Au-Col-FITC in MSC significantly decreased with CPZ and BAF inhibitors, indicating that clathrin-mediated endocytosis and vacuolar-type H^+^-ATPase pathway are the major mechanisms whereby MSC “eats” AuNP-Col-FITC. Regarding the diameter of AuNP-Col-FITC, which was measured as 255.96 nm (Figure 1D), a greater uptake amount was seen after 24 h in Wharton’s jelly MSCs. In the literature, it has been shown that nanoparticles with a diameter of 10–300 nm are taken up by cells via clathrin-mediated endocytosis. Our results support this finding [49].

Lysosome is an organelle responsible for intracellular transportation after uptake of foreign substrates into cells by mechanisms such as phagocytosis and endocytosis to maintain cellular homeostasis. The microenvironment in the lysosome is acidic (pH 4.5–5.0) and lysosomes recruit enzymes to degrade foreign molecules [50]. Thus, the stability of the AuNP delivery system in Wharton’s jelly MSCs was further explored in this study. According to the results of the LysoTracker assay in the present study, it was demonstrated AuNP-Col-FITC was not co-localized with lysosomes, which indicates it had good stability that could not be degraded and could escape the influence of lysosomes (Figure 9).

Currently, nanotechnology, including metallic particles, liposomes, micelles, hydrogels and polymeric nanoparticles, are used in biomedical approaches [51]. However, the drawbacks, such as cytotoxicity, induction of inflammatory responses, limited targeting efficiency and being expensive to develop need to be overcome [51]. The titanium (Ti) nanoparticles have been verified to induce inflammation and DNA damage leading to cell apoptosis in various cell lines [52,53]. Moreover, the titanium dioxide (TiO_2_) nanoparticles were reported to cause injury in organs such as the brain and liver through oxidative DNA damage [54,55]. Meanwhile, porous silica nanoparticles have been studied for medical uses, but the silanol groups on the surface of porous silica nanoparticles could interact with the phospholipids of erythrocytes to cause hemolysis [56]. Polymeric nanoparticles, such as PLGA-based or PCL-based delivery systems, have been reported to be hydrophobic and need longer degradation time [57]. As mentioned above, AuNPs are considered a promising nanodrug delivery platform. When combined with MSC therapies, the advantages of AuNPs are amplified. According to recent research, AuNPs offer several advantages in drug delivery, such as enhanced therapeutic efficacy, improved bioavailability and targeted delivery [58,59,60]. The high surface area to volume ratio of AuNPs allows for increased drug loading capacity, leading to improved therapeutic efficacy [30]. Additionally, AuNPs can be functionalized with targeting molecules, such as antibodies or aptamers, allowing for targeted drug delivery to specific cell types or tissues [61]. Additionally, our studies have verified that AuNPs modified with collagen are a biocompatible nanodrug delivery system that can carry natural compounds and has demonstrated good anti-cancer capacities [18,62]. However, since the higher concentration of AuNP may lead to cytotoxicity, the mission to select appropriate concentration of nanoparticles to develop a nanodrug delivery system becomes important for regenerative therapies [40].

Stem cells, including neural stem cells (NSCs), embryonic stem cells (ESCs) and adipose-derived stem cells (ASCs) are widely used for clinical therapies in neurodegenerative diseases and other conditions [51]. However, several disadvantages, such as being difficult to obtain, potential for tumor formation and lack of capability for self-renewal, complicate their use [63,64]. Wharton’s jelly MSCs have several advantages as candidates for stem cell therapy owing to the higher expression of Oct4 and Sox2 pluripotency markers and anti-inflammatory abilities [65]. Therefore, when coupled with MSCs, AuNPs can be used as vehicles to transport drugs to sites of inflammation or disease, thereby improving their bioavailability and reducing off-target effects [66,67]. MSCs can also provide protection for AuNPs against clearance by the immune system, prolonging their circulation time to facilitate therapeutic efficacy [68]. This study found that AuNP at concentrations of 1.25 and 2.5 ppm could facilitate proliferation and migration ability by activating the SDF-1α/CXCR4 pathway. The stability of AuNP-Col shown in this study therefore indicates that it could be a promising nanoparticle for future therapies. Overall, the combination of AuNPs and Wharton’s jelly MSCs offers a promising approach for drug delivery with a number of advantages.

## 5. Conclusions

In the present research, we knocked down the CXCR4 expression in Wharton’s jelly MSCs to evaluate the biological performance induced by AuNP. After the transfection of CXCR4 siRNA into Wharton’s jelly MSCs, the cell viability was suppressed, and MMP expression was also inhibited. However, after treatment with 1.25 ppm and 2.5 ppm of AuNP solution, the biological performances, including expression of CXCR4 and MMP-2/9 proteins, were significantly enhanced, which positively facilitated the migration ability of MSCs by activating the SDF-1α/CXCR4 signal pathway. Additionally, it was verified that MSC could uptake AuNP through the clathrin-mediated endocytosis and vacuolar-type H^+^-ATPase pathways, and AuNP showed good stability in cells. In conclusion, AuNPs, at appropriate concentrations of 1.25 and 2.5 ppm, demonstrated good efficiency in strengthening the biological performances of MSCs and enhanced biocompatibility in animal models; taken together, this indicates that a combination of AuNP and Wharton’s jelly MSCs could serve as a nanodrug delivery system with tissue regenerative applications.

## Figures and Tables

**Figure 1 pharmaceutics-15-01385-f001:**
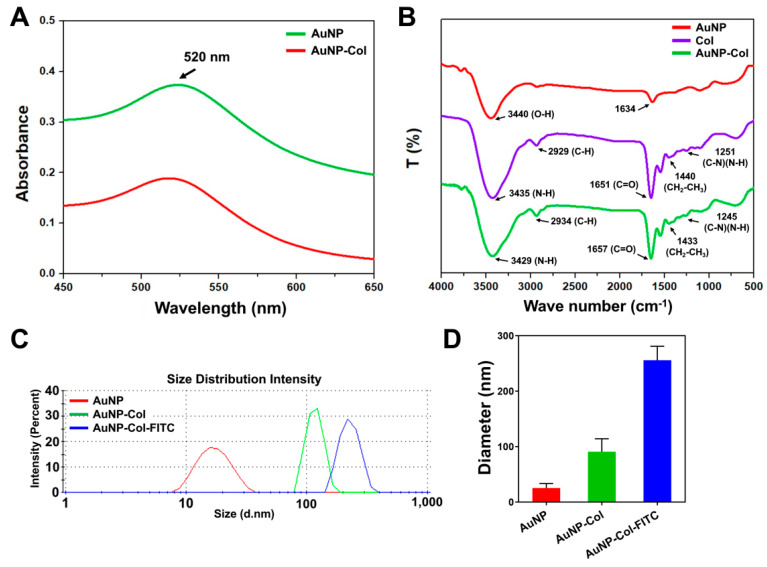
Identification of AuNP-derived nanomaterials. (**A**) The UV-Vis spectrum indicated the typical peak of AuNP at the wavenumber of 520 nm in AuNP-Col-FITC, which demonstrated the presence of AuNP. (**B**) The FTIR spectrum of pure AuNP, pure Col, and AuNP-Col nanomaterials. (**C**,**D**) The size distribution intensity and diameter of nanoparticles were measured by DLS assay. The result is displayed as mean ± SD (*n* = 3).

**Figure 2 pharmaceutics-15-01385-f002:**
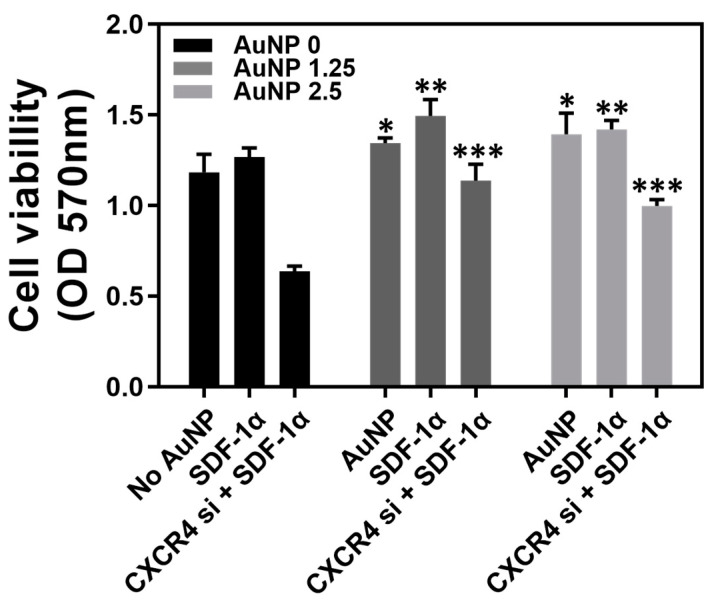
Cell viability of Wharton’s jelly MSCs with various treatments after 48 h incubation. The MSCs were treated with different treatments: AuNP 0 ppm group (No AuNP, SDF-1α, CXCR4 siRNA + SDF-1α), AuNP 1.25 ppm (pure AuNP, SDF-1α, CXCR4 siRNA + SDF-1α), and AuNP 2.5 ppm (pure AuNP, SDF-1α, CXCR4 siRNA + SDF-1α). The results were quantified and compared with the AuNP 0 ppm group in triplicate. * *p* < 0.05, ** *p* < 0.01, *** *p* < 0.001: compared to AuNP 0 ppm group.

**Figure 3 pharmaceutics-15-01385-f003:**
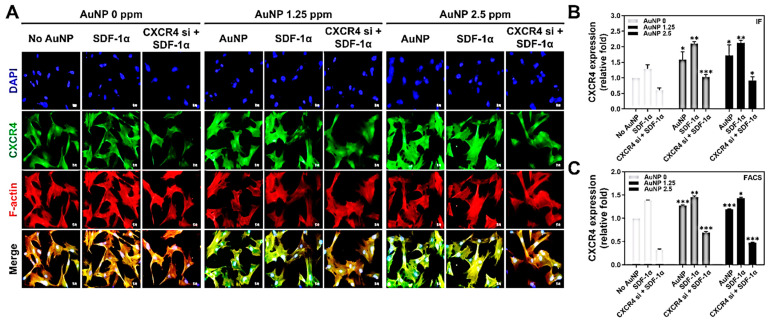
The CXCR4 expression intensity after 48 h of AuNP treatments. The Wharton’s jelly MSCs were treated with various treatments (SDF-1α, CXCR4 siRNA + SDF-1α) prior to the addition of 0 ppm, 1.25 ppm and 2.5 ppm AuNP, to further investigate the CXCR4 expression through immunofluorescence (IF) and FACS methods. (**A**) The images of CXCR4 expression were captured with a fluorescence microscope. Green color: CXCR4 expression intensity; red color: F-actin; blue color: cell nuclear staining by DAPI. Scale bars: 20 μm. (**B**) The quantitative results of CXCR4 expression based on fluorescence intensity. (**C**) The fluorescein-positive cells were detected by flow cytometry and analyzed with the FACS method. The results were quantified and compared with the AuNP 0 ppm group in triplicate. * *p* < 0.05, ** *p* < 0.01, *** *p* < 0.001: compared to AuNP 0 ppm group.

**Figure 4 pharmaceutics-15-01385-f004:**
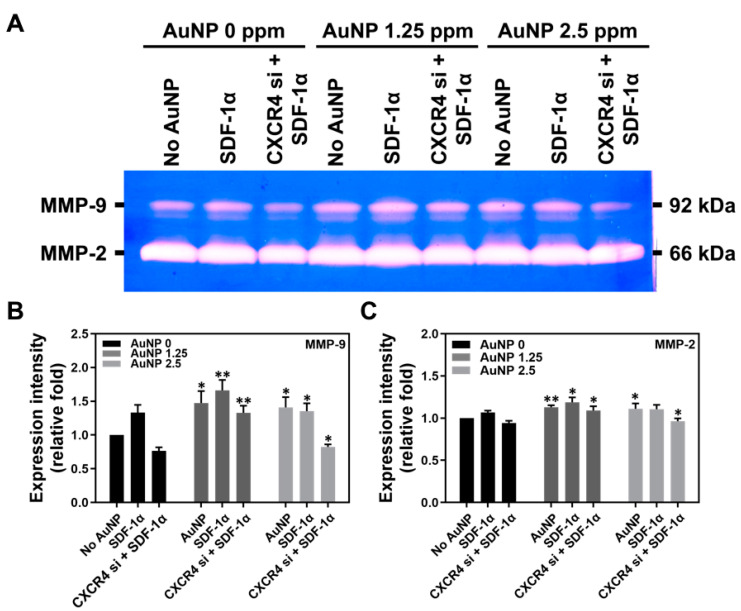
The expression of matrix metalloproteinases (MMPs) in Wharton’s jelly MSCs induced by various concentrations of AuNP after 48 h. (**A**) The zymogram of MMP-9 and MMP-2 expression in MSCs are shown. The expression of (**B**) MMP-9 and (**C**) MMP-2 was quantified by Image J 5.0 software based on the expression intensity. The results were quantified and compared with the AuNP 0 ppm group in triplicate. * *p* < 0.05, ** *p* < 0.01: compared to AuNP 0 ppm group.

**Figure 5 pharmaceutics-15-01385-f005:**
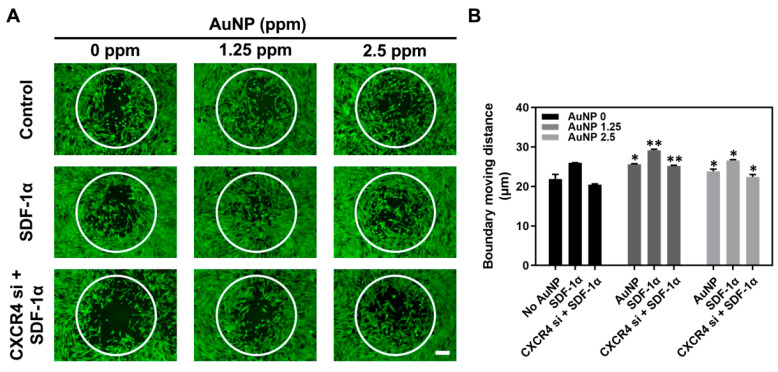
Wharton’s jelly MSCs migration ability with AuNP treatment for 48 h. The MSCs were assigned to three groups: No treatment, SDF-1α treatment, and CXCR4 si + SDF-1α treatment. In addition, AuNP 0 ppm, 1.25 ppm and 2.5 ppm were added to the above three groups to observe the induction of migration ability. (**A**) The movement of MSCs into the boundary area was observed by fluorescence intensity within each group. The scale bars are equal to 200 μm. (**B**) The distance of MSC movement in the boundary area was further quantified, and the results were compared with the AuNP 0 ppm group in triplicate. * *p* < 0.05, ** *p* < 0.01: compared to AuNP 0 ppm group.

**Figure 6 pharmaceutics-15-01385-f006:**
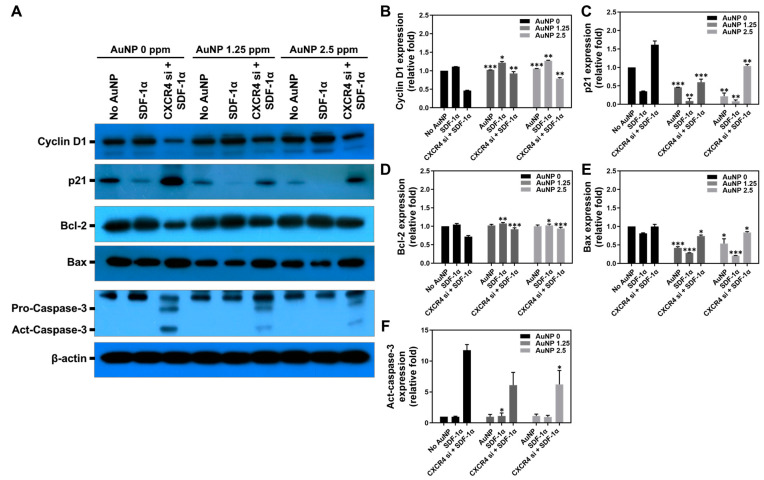
Expression of cell apoptotic-related proteins in Wharton’s jelly MSCs with various treatments for 48 h. The MSCs were assigned to three groups: No treatment, SDF-1α treatment and CXCR4 si + SDF-1α treatment. Next, the different concentrations of AuNP solution were added for investigation of cell apoptosis. (**A**) The immunoblots of apoptotic-related proteins expressed in MSC are depicted. The expression of each protein was quantified as: (**B**) Cyclin D1, (**C**) p21, (**D**) Bcl-2, (**E**) Bax and (**F**) Act-Caspase-3. The results were compared with the AuNP 0 ppm group in triplicate. * *p* < 0.05, ** *p* < 0.01, *** *p* < 0.001: compared to AuNP 0 ppm group.

**Figure 7 pharmaceutics-15-01385-f007:**
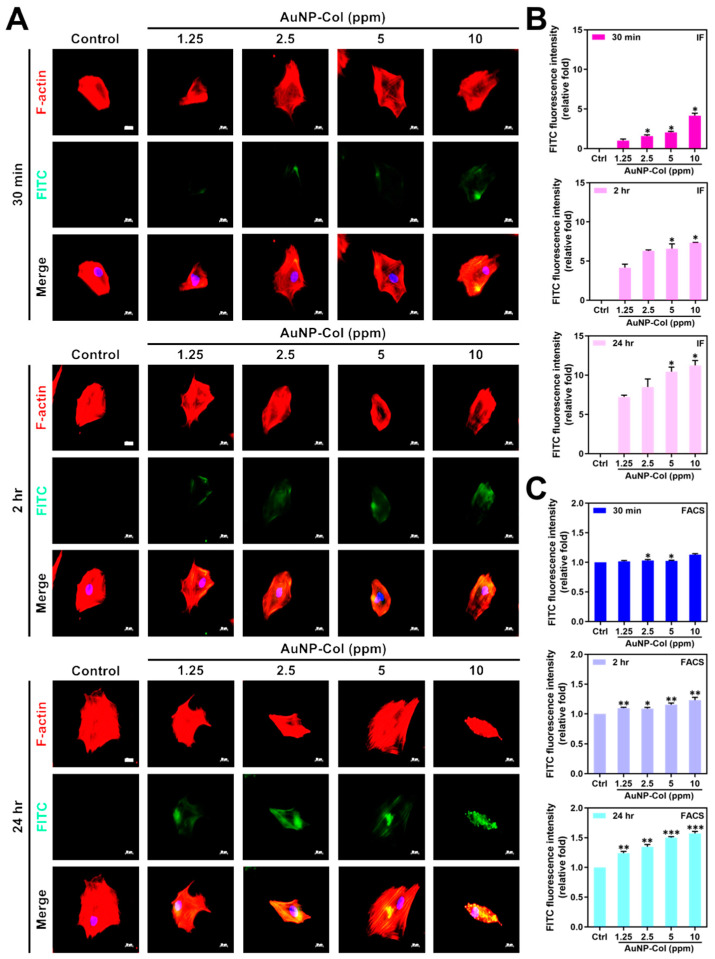
The AuNP-Col uptake efficiency in Wharton’s jelly MSCs at 30 min, 2 h and 24 h. Various concentrations of AuNP were combined with Type I collagen, and the resultant compound was named AuNP-Col. Further conjugation with FITC was performed (green fluorescence). (**A**) The fluorescence images were captured by fluorescence microscopy, and the scale bars are equal to 20 μm. Cell nuclei were located by DAPI (blue color), and F-actin was stained by rhodamine phalloidin (red color). (**B**,**C**) The MSC uptake efficiency was determined by both IF and FACS methods. The results were quantified in triplicate. * *p* < 0.05: compared to the AuNP 1.25 ppm group in IF method. * *p* < 0.05, ** *p* < 0.01, *** *p* < 0.001: compared to the control group in the FACS method (treatment without AuNP-Col).

**Figure 8 pharmaceutics-15-01385-f008:**
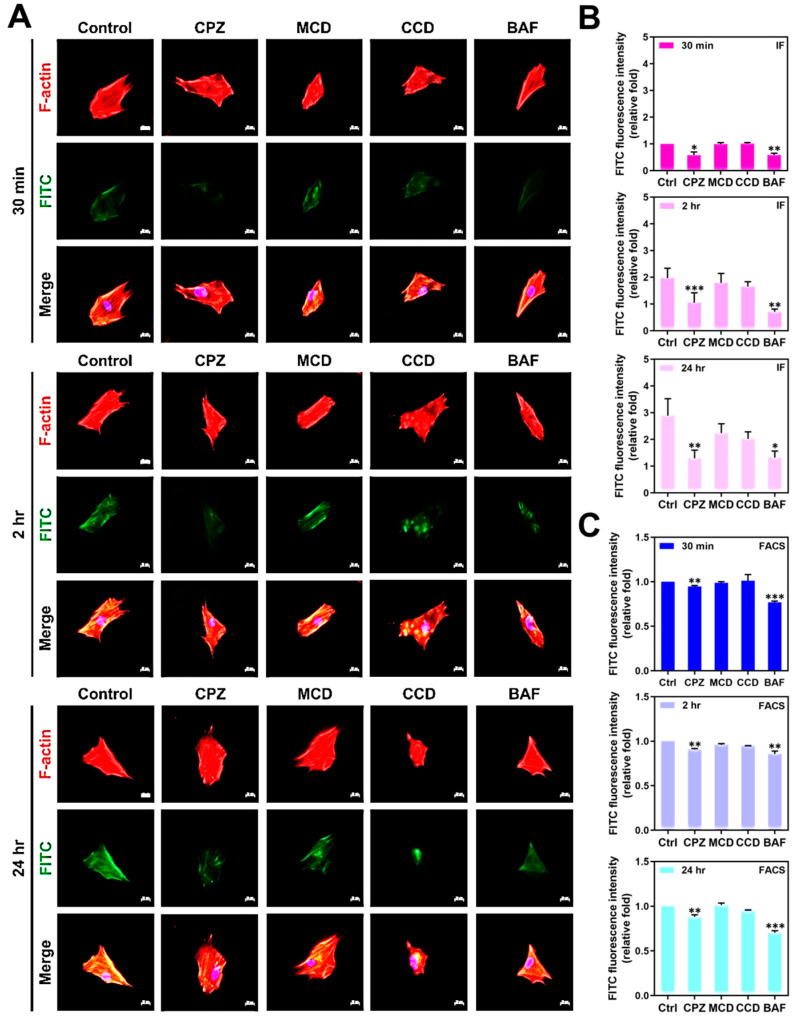
The endocytosis mechanisms of MSC uptake AuNP-Col (2.5 ppm) investigated by various inhibitors. (**A**) The MSCs were pre-treated with different inhibitors. Next, the AuNP-Col-FITC 2.5 ppm solutions were added and incubated for 30 min, 2 h and 24 h. The images were obtained by fluorescence microscopy. Scale bar: 20 μm. Green fluorescence: AuNP-Col-FITC (2.5 ppm); red fluorescence: F-actin; blue fluorescence: cell nuclear stained by DAPI. (**B**,**C**) The uptake amount quantified by FITC fluorescence intensity was analyzed based on IF and FACS methods. The results were quantified in triplicate. * *p* < 0.05, ** *p* < 0.01, *** *p* < 0.001: compared to the control (treatment without inhibitors).

**Figure 9 pharmaceutics-15-01385-f009:**
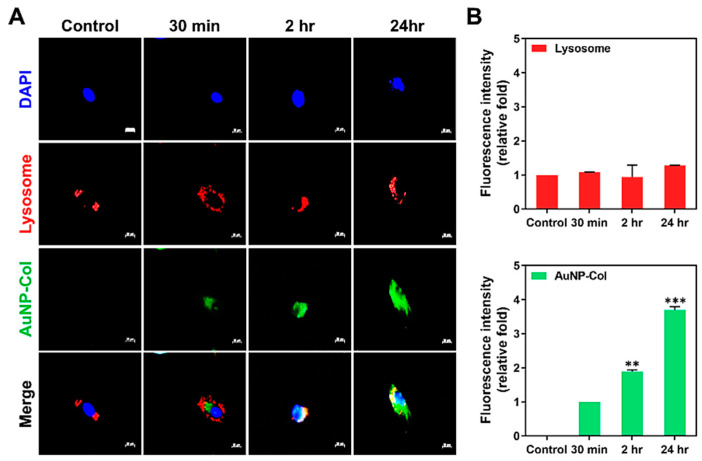
The intracellular transportation of AuNP-Col in Wharton’s jelly MSCs investigated by LysoTracker assay at 30 min, 2 h and 24 h. (**A**) The fluorescence images were acquired by fluorescence microscopy. Scale bar: 20 μm. Green color: AuNP-Col-FITC (2.5 ppm); red color: lysosome; blue color: cell nuclei were stained by DAPI solution. (**B**) The fluorescence intensity of lysosome and AuNP-Col-FITC were analyzed and quantified by Image J software. The results were quantified in triplicate. ** *p* < 0.01, *** *p* < 0.001: compared to the 30 min group.

**Figure 10 pharmaceutics-15-01385-f010:**
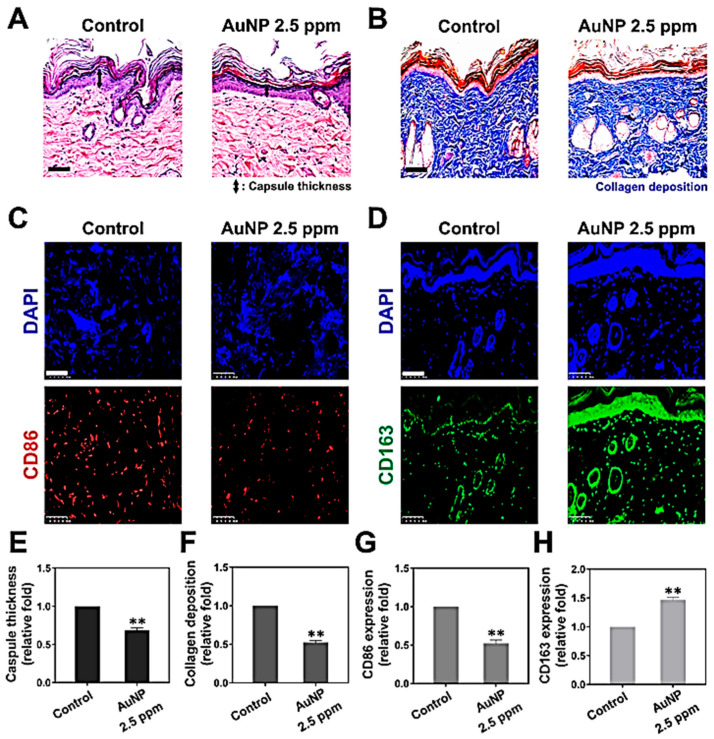
The foreign body response induced by AuNP 2.5 ppm by in vivo animal models. (**A**) The capsule formation was demonstrated by H&E staining. (**B**) The collagen deposition was evaluated by Masson’s trichrome staining assay. (**C**,**D**) The M1 (CD86, red color) and M2 (CD163, green color) macrophage polarization was investigated by immunohistochemistry staining assay. DAPI was used for cell nuclei staining (blue color). The scale bars were 100 μm. The results were quantified as: (**E**) capsule thickness, (**F**) collagen deposition, (**G**) CD86 expression and (**H**) CD163 expression. The results were quantified in triplicate. ** *p* < 0.01: compared to the control (treatment without AuNP 2.5 ppm).

**Figure 11 pharmaceutics-15-01385-f011:**
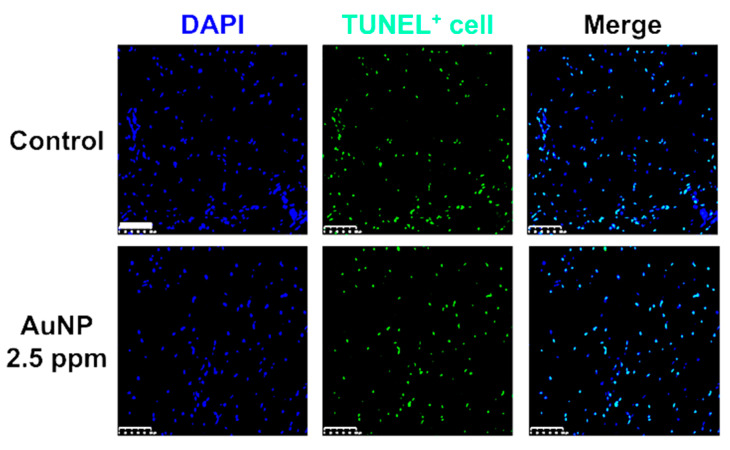
Detection of apoptotic cells in animal tissue via TUNEL assay. The green fluorescence cells indicated TUNEL-positive cells as well as apoptotic cells in animal tissue. The statistical analysis indicated no significant difference between treatment groups and the control group, which demonstrated that AuNP was a biosafe nanomaterial.

**Figure 12 pharmaceutics-15-01385-f012:**
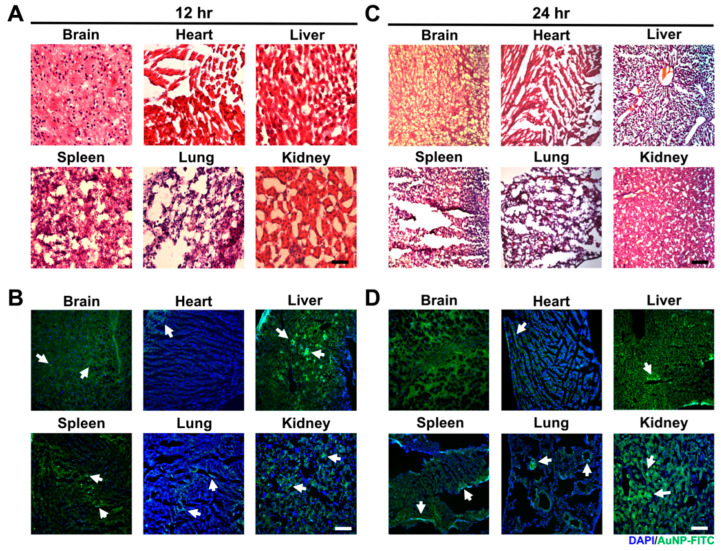
Examination of tissue integrity and particle distribution after AuNP treatments for 12 and 24 h. AuNP solution at a concentration of 2.5 ppm was injected into the retro-orbital sinus of the experimental mice. The brain, heart, liver, spleen, lung and kidney tissues, were subjected to H&E staining assay and fluorescence observation. (**A**,**B**) The images of tissue morphology and particle distribution were exhibited after 12 hours’ treatment. (**C**,**D**) The images of tissue morphology and particle distribution after 24 h treatment. The results of H&E staining demonstrated AuNP did not cause serious destruction in any of the tissue types, and the nanoparticles (green fluorescence) could be observed in these tissues. This indicates that AuNP may have potential for nanodrug applications. DAPI was used to observe the cell nuclei. Scale bar: 50 μm. Arrow: FITC-positive cells.

**Figure 13 pharmaceutics-15-01385-f013:**
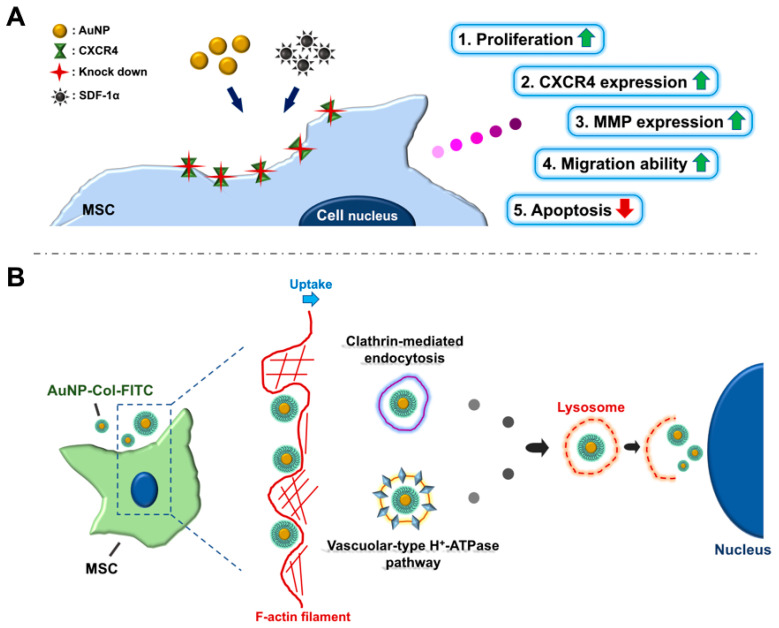
(**A**) The illustration indicates that AuNP at concentrations of 1.25 and 2.5 ppm could strengthen the Wharton’s jelly MSC biological performance via the SDF-1α/CXCR4 pathway. (**B**) A schematic diagram of the endocytic pathways of AuNP-Col uptake by Wharton’s jelly MSCs.

**Table 1 pharmaceutics-15-01385-t001:** Cyclin D1 expression level in Wharton’s jelly MSCs.

Cyclin D1 (Relative Fold)	AuNP	SDF-1α	CXCR4 si + SDF-1α
AuNP 0 ppm	1.00	1.11	0.46
AuNP 1.25 ppm	1.02 ***	1.21 *	0.92 **
AuNP 2.5 ppm	1.05 ***	1.26 **	0.79 **

Note: * *p* < 0.05, ** *p* < 0.01, *** *p* < 0.001: compared with AuNP 0 ppm treatment.

**Table 2 pharmaceutics-15-01385-t002:** p21 expression level in Wharton’s jelly MSCs.

p21 (Relative Fold)	AuNP	SDF-1α	CXCR4 si + SDF-1α
AuNP 0 ppm	1.00	0.35	1.61
AuNP 1.25 ppm	0.46 ***	0.09 **	0.59 ***
AuNP 2.5 ppm	0.21 **	0.08 **	1.03 **

Note: ** *p* < 0.01, *** *p* < 0.001: compared with AuNP 0 ppm treatment.

**Table 3 pharmaceutics-15-01385-t003:** Bcl-2 expression level in Wharton’s jelly MSCs.

Bcl-2 (Relative Fold)	AuNP	SDF-1α	CXCR4 si + SDF-1α
AuNP 0 ppm	1.00	1.04	0.71
AuNP 1.25 ppm	1.02	1.07 **	0.92 ***
AuNP 2.5 ppm	1.00	1.02 *	0.94 ***

Note: * *p* < 0.05, ** *p* < 0.01, *** *p* < 0.001: compared with AuNP 0 ppm treatment.

**Table 4 pharmaceutics-15-01385-t004:** Bax expression level in Wharton’s jelly MSCs.

Bax (Relative Fold)	AuNP	SDF-1α	CXCR4 si + SDF-1α
AuNP 0 ppm	1.00	0.81	0.99
AuNP 1.25 ppm	0.42 ***	0.28 ***	0.74 *
AuNP 2.5 ppm	0.53 *	0.20 ***	0.83 *

Note: * *p* < 0.05, *** *p* < 0.001: compared with AuNP 0 ppm treatment.

**Table 5 pharmaceutics-15-01385-t005:** Active-Caspase-3 expression level in Wharton’s jelly MSCs.

Active-Caspase-3 (Relative Fold)	AuNP	SDF-1α	CXCR4 si + SDF-1α
AuNP 0 ppm	1.00	0.98	11.75
AuNP 1.25 ppm	0.99	1.11 *	6.10
AuNP 2.5 ppm	1.13	0.95	6.22 *

Note: * *p* < 0.05: compared with AuNP 0 ppm treatment.

## Data Availability

Data are contained within the article.

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
