# Peer review of "Therapeutic Applications of Mesenchymal Stem Cell Loaded with Gold Nanoparticles for Regenerative Medicine"

_pharmaceutics, 2023, doi:10.3390/pharmaceutics15051385_

Round 1

Reviewer 1 Report

In this manuscript the group investigated the relationship between gold nanoparticles and mesenchymal stem cells. The study was well presented compromise of in vitro and in vivo experiments. Some minor changes:

- Better to describe the type of MSC used in the context of the manscript – Wharton’s MSCs.

- Include more information for the cell culture of MSC. How were these MSC selected and verified to be MSCs?

Author Response

Reviewer 1:

In this manuscript the group investigated the relationship between gold nanoparticles and mesenchymal stem cells. The study was well presented compromise of in vitro and in vivo experiments. Some minor changes:

-Better to describe the type of MSC used in the context of the manuscript – Wharton’s MSCs.

Answer:
Thanks for the valuable comment from the Reviewer. We have revised for the description of MSCs as Wharton’s jelly MSCs in the manuscript.

  1. Include more information for the cell culture of MSC. How were these MSC selected and verified to be MSCs?

Answer:
Thanks for the suggestion from the Reviewer. We have included the detail description for identification of Wharton’s jelly mesenchymal stem cells in the “Section 2.2” and in the “Section 3.2”. The verify result and flow histograms are added in the new “Supplementary figure S1”.

(1) “The human umbilical cord Wharton’s jelly mesenchymal stem cells (MSCs) were applied for the present research, which were also selected in our previous literatures [9, 34]. The cells were stored cultured in H-DMEM medium (Invitrogen, USA) containing 10% fetal bovine serum (FBS) and 1% (v/v) antibiotics (100 U/mL penicillin/streptomycin). The cells at the 8th passage used for various experiments were cautiously cultured in a 37°C incubator with 5% CO2 humidified atmosphere.

To identify the phenotypes of Wharton’s jelly MSCs, the cells were detached by 2mM EDTA with PBS. The MSCs were washed by PBS containing 2% bovine serum albumin (BSA) and 0.1% sodium azide (Sigma, USA). Next, the cells were cultured with various specific antibodies including CD14-FITC-A, CD34-FITC-A, CD45-FITC-A, CD29-FITC-A, CD44-PE-A and CD73-PE-A (fluorescein isothiocyanate represented FITC and phycoerythrin denoted as PE). The FITC/PE conjugated IgG1 were used for isotype controls (BD Pharmingen, USA). A flow cytometer was applied for MSC phenotype detection.” (Page 4, Line 158-170)

(2) “The phenotypes of Wharton’s jelly MSCs used in this study were identified by CD14, CD29, CD34, CD44, CD45 and CD73 surface markers. Figure S1 displayed the flow cytometric results of each specific marker expression. The expression of CD14, CD34 and CD45 endothelial markers was analyzed as 0.43%, 0.81% and 0.335% (negative markers).  The cells expressed CD29, CD44 and CD73 positive markers as 97.7%, 93.6% and CD73%, which demonstrated the expression of MSC surface markers.” (Page 8, Line 375-380)
(3) “Figure S1. Phenotype identification of Wharton’s jelly MSCs. The surface specific markers expression in cells were detected by flow cytometry. The expression of CD14, CD34 and CD45 negative markers were analyzed as 0.43%, 0.81 % and 0.335 %. And the expression of CD29, CD44 and CD73 positive markers was 97.7%, 93.6% and 95.9% in cells, respectively. The above data represents of three independent experiments.” (Supplementary figure S1)

References:

(9) Chang, K.-B.; Shen, C.-C.; Hsu, S.-h.; Tang, C.M.; Yang, Y.-C.; Liu, S.-Y.; Ku, T.R.; Kung, M.-L.; Hsieh, H.-H.; Hung, H.-S. Functionalized collagen-silver nanocomposites for evaluation of the biocompatibility and vascular differentiation capacity of mesenchymal stem cells. Colloids and Surfaces A: Physicochemical and Engineering Aspects 2021, 624, 126814.

(34) Hung H-S, Yang Y-C, Chang C-H, et al. Neural differentiation potential of mesenchymal stem cells enhanced by biocompatible chitosan-gold nanocomposites. Cells 2022;11:1861.

Reviewer 2 Report

Dear author’s

I was pleased to review your manuscript « Therapeutic Applications of Mesenchymal Stem Cell Loaded with Gold Nanoparticles for Regenerative Medicine ». The study is in the scope of Journal Pharmaceutics. The subject is quite interesting.  The experiments seem to be carried out carefully and the results are clearly presented. Overall, the paper is well written with good English. However, the following issues raised should be addressed before publication.

1.      Abstract section looks incomplete and not properly arrange. I will suggest the author to focus on following important points on writing the abstract. An abstract summarizes, usually in one paragraph of 300 words or less, the major aspects of the entire paper in a prescribed sequence that includes: 1) the overall purpose of the study and the research problem(s) you investigated; 2) the basic design of the study; 3) major findings or trends found as a result of your analysis; and, 4) a brief summary of your interpretations and conclusions.

2.      Lines 66-67: The authors' conclusion that "the chemical process used to produce AuNP produces residues of chemical reagents that may induce cytotoxicity and side effects" seems unfounded, especially when sodium citrate is used. Furthermore, gold nanoparticles can be produced by a variety of methods. Why did the authors stop at the reference [12]?

3.      In fact, the originality and novelty of your paper should be better highlighted in relation to the current state of knowledge.

4.      In my opinion, it is necessary to provide TEM images of statistically significant amounts of both types of nanoparticles (AuNP and AuNP-Col), especially since the authors claim that nanoparticle size has a significant influence on their bioavailability. The optical absorption spectra of gold nanoparticles presented by the authors contain a very broad absorption band, indicating the likely polydispersity of the particles.

5.      Authors claim, that «gold nanoparticle solution was free from chemical compounds», however Figure 1B which shows the FTIR spectra of AuNP does not confirm it. The spectrum clearly shows absorption bands corresponding not only to water and OH groups. Please, explain.

6.      Please check the significant figures in the numbers along all research based on a confidence interval for the mean.

7.      How to explain that the viability of MSCs was enhanced by pure AuNP 1.25 ppm and AuNP 2.5 ppm? Why does the expression intensity decrease as the concentration of Au NPs increases? What are the reasons for these phenomena?

8.      The discussion of the results is poor. In fact, the Discussion section contains a summary of the results. Besides, any comparison of the obtained data with the literature is lack. The discussion section should be improved. An improvement of the discussion is needed also in relation to similar or analogous results from literature and the explanation of possible differences.

Author Response

Reviewer 2:

Dear author’s

I was pleased to review your manuscript « Therapeutic Applications of Mesenchymal Stem Cell Loaded with Gold Nanoparticles for Regenerative Medicine ». The study is in the scope of Journal Pharmaceutics. The subject is quite interesting.  The experiments seem to be carried out carefully and the results are clearly presented. Overall, the paper is well written with good English. However, the following issues raised should be addressed before publication. 

  1. Abstract section looks incomplete and not properly arrange. I will suggest the author to focus on following important points on writing the abstract. An abstract summarizes, usually in one paragraph of 300 words or less, the major aspects of the entire paper in a prescribed sequence that includes: 1) the overall purpose of the study and the research problem(s) you investigated; 2) the basic design of the study; 3) major findings or trends found as a result of your analysis; and, 4) a brief summary of your interpretations and conclusions.

Answer:
Thanks for the kindly suggestion from the Reviewer. We have revised the description in the “Abstract” section. (Page 1, Line 20-37)
“In the present study, the various concentrations of AuNP (1.25, 2.5, 5, 10 ppm) were prepared to investigate the biocompatibility, biological performances and cell uptake efficiency via Wharton’s jelly mesenchymal stem cells and rat model. The pure AuNP, AuNP combined with Col (AuNP-Col) and FITC conjugated AuNP-Col (AuNP-Col-FITC) were characterized by Ultraviolet–visible spectroscopy (UV-Vis), Fourier-transform infrared spectroscopy (FTIR) and Dynamic Light Scattering (DLS) assays. For in vitro examinations, we explored that the Wharton’s jelly MSCs had better viability, higher CXCR4 expression, greater migration distance and lower apoptotic-related proteins expression with AuNP 1.25 and 2.5 ppm treatments. Furthermore, the treatments of 1.25 and 2.5 ppm AuNP could induce the CXCR4 knocked down Wharton’s jelly MSCs to express CXCR4 and reduce the expression level of apoptotic proteins. We also treated the Wharton’s jelly MSCs with AuNP-Col for the investigations of intracellular uptake mechanisms. The evidence demonstrated the cells uptake AuNP-Col through clathrin-mediated endocytosis and the vacuolar-type H+-ATPase pathway with good stability inside the cells to avoid lysosomal degradation as well as better uptake efficiency. Additionally, the results from in vivo examinations elucidated the 2.5 ppm of AuNP attenuated foreign body responses and had better retention efficacy with tissue integrity in animal model. In conclusion, the evidence demonstrates the AuNP shows promise as a biosafe nanodrug delivery system for development of regenerative medicine coupled with Wharton’s jelly MSCs.”

  1. Lines 66-67: The authors' conclusion that "the chemical process used to produce AuNP produces residues of chemical reagents that may induce cytotoxicity and side effects" seems unfounded, especially when sodium citrate is used. Furthermore, gold nanoparticles can be produced by a variety of methods. Why did the authors stop at the reference [12]?

Answer:
Thanks for the comment. We have included new description in Introduction section. (Page 2, Line 66-71)
“The AuNP obtained from chemical methods is often conducted by using tetrachloroauric acid with sodium citrate, sodium borohydride or other toxic reagents as reducing and stabilizing agents for the synthesis process [12, 13]. A study reported that the nanoparticles synthesized via green approaches demonstrated lower cytotoxicity than chemically method [14]. Therefore, the eco-friendly methods can be suggested as the promising approach to obtain nanoparticles [15].”

References:

(12) Sarfraz, N.; Khan, I. Plasmonic gold nanoparticles (AuNPs): properties, synthesis and their advanced energy, environmental and biomedical applications. Chemistry–An Asian Journal 2021, 16, 720-742.

(13) da Silva Moraes1a, D.; Biza, H.M.; de Campos, T.L.A. GOLD NANOPARTICLES SYNTHESIS WITH DIFFERENT REDUCING AGENTS CHARACTERIZED BY UV-VISIBLE ESPECTROSCOPY AND FTIR.

(14) Darvishi, E.; Kahrizi, D.; Arkan, E. Comparison of different properties of zinc oxide nanoparticles synthesized by the green (using Juglans regia L. leaf extract) and chemical methods. Journal of Molecular Liquids 2019, 286, 110831.

(15) Yazdanian, M.; Rostamzadeh, P.; Rahbar, M.; Alam, M.; Abbasi, K.; Tahmasebi, E.; Tebyaniyan, H.; Ranjbar, R.; Seifalian, A.; Yazdanian, A. The Potential Application of Green-Synthesized Metal Nanoparticles in Dentistry: A Comprehensive Review. Bioinorganic Chemistry and Applications 2022, 2022.

  1. In fact, the originality and novelty of your paper should be better highlighted in relation to the current state of knowledge.

Answer:

Thanks for the valuable suggestion. We have included new descriptions in the “Discussion” section and in the “Conclusion” sections.

(1) “Current state of nanotechnology including metallic particles, liposomes, micelles, hydrogels and polymeric nanoparticles are applied for biomedical approaches [51]. However, the drawbacks such as cytotoxicity, induction of inflammatory responses, limited targeting efficiency and expensive for development need to be overcome [51]. The titanium (Ti) nanoparticles have been verified to induce inflammations and DNA damage leading to cell apoptosis in various cell lines [52, 53]. Moreover, the titanium dioxide (TiO2) nanoparticles were reported to cause injury in organs such as brain and liver due to oxidative DNA damage [54, 55]. Meanwhile, the porous silica nanoparticles have been studied for medical approaches, but the silanol groups on the surface of porous silica nanoparticles could interact with the phospholipids of erythrocyte to cause hemolysis [56]. For the polymeric nanoparticles such as PLGA-based or PCL-based delivery system, they have been reported to be hydrophobicity and need longer degradation time [57].” (Page 20, Line 708-719)

(2) “Stem cells are widely applied for clinical therapies such as neurodegenerative diseases, including neural stem cells (NSCs), embryonic stem cells (ESCs) and adipose-derived stem cells (ASCs) [64]. However, several disadvantages such as difficult to obtain, potential of tumor formation and lack of self-renewal ability [65, 66]. Wharton’s jelly MSCs have several advantages to be the candidate for stem cell therapy owing to the higher expression of Oct4 and Sox2 pluripotency markers and anti-inflammatory abilities [67].” (Page 21, Line 733-738)

(3) “In conclusion, AuNP at appropriate concentrations of 1.25 and 2.5 ppm demonstrated good efficiency in strengthening the biological performances of MSCs and enhanced biocompatibility in animal models, which taken together, indicate that a combination of AuNP and Wharton’s jelly MSCs could serve as a nanodrug delivery system with tissue regenerative applications.” (Page 21, Line 757-761)

  1. In my opinion, it is necessary to provide TEM images of statistically significant amounts of both types of nanoparticles (AuNP and AuNP-Col), especially since the authors claim that nanoparticle size has a significant influence on their bioavailability. The optical absorption spectra of gold nanoparticles presented by the authors contain a very broad absorption band, indicating the likely polydispersity of the particles.

Answer:
Thanks for the valuable suggestion. We agree with the comment from the Reviewer. However, due to the relocation of laboratory to a new research building, the TEM instrument is still not available for operation. Nevertheless, the TEM images, zeta-potential, zeta-deviation and PDI value related to AuNP and AuNP-Col can be referenced in our previous literature, while the results are demonstrated as below. The new description and citation are included in the “Results” section.
“The TEM images for Au and AuNP-Col could be referenced in our published study, while the polydispersity index was also included [37].” (Page 8, Line 365-367)

Reference:

  1. Chen, H.-C.; Kung, M.-L.; Huang, W.-X.; Fu, R.-H.; Yu, A.Y.-H.; Yang, Y.-T.; Hung, H.-S. Delivery of stromal-derived factor-1α via biocompatible gold nanoparticles promotes dendritic cells viability and migration. Colloids and Surfaces A: Physicochemical and Engineering Aspects 2021, 628, 127298.
  2. Authors claim, that «gold nanoparticle solution was free from chemical compounds», however Figure 1B which shows the FTIR spectra of AuNP does not confirm it. The spectrum clearly shows absorption bands corresponding not only to water and OH groups. Please, explain.

Answer:
Thanks for the comment from the Reviewer. We have included the more detail description in the “Results” section (Page 7, Line 355-361)

“According to the previous literature published in 1989, the IR measurement from liquid water demonstrated the bands at around 3400 cm-1 (symmetric and antisymmetric stretch of O-H bonding) and 1648 cm-1 (H-O-H bending) [35]. Another study published in 2004 showed the FTIR results of AuNP produced in deionized water, which reported the stretch bonding of carbonato complexes (Au-OCO2- and Au-OCO2H) on AuNP at 1000, 1334 and 1595 cm-1 [36] due to the atmospheric CO2 is slightly soluble in water to form HCO3- and H+ during the preparation of samples.”

References:

(35) Malhotra, V.; Jasty, S.; Mu, R. FT-IR spectra of water in microporous KBr pellets and water's desorption kinetics. Applied spectroscopy 1989, 43, 638-645.

(36) Sylvestre, J.-P.; Poulin, S.; Kabashin, A.V.; Sacher, E.; Meunier, M.; Luong, J.H. Surface chemistry of gold nanoparticles produced by laser ablation in aqueous media. The Journal of Physical Chemistry B 2004, 108, 16864-16869.

  1. Please check the significant figures in the numbers along all research based on a confidence interval for the mean.

Answer:
Thanks for the suggestion from the Reviewer. We have checked the figures contained p value investigations for each semi-quantitative result in present study.

Figure 3. Cell viability of Wharton’s jelly MSCs with various treatments after 48 hours’ incubation. *p < 0.05, **p < 0.01, ***p < 0.001: compared to AuNP 0 ppm group.

Figure 4BC. The CXCR4 expression intensity after 48 hours of AuNP treatments. *p < 0.05, **p < 0.01, ***p < 0.001: compared to AuNP 0 ppm group.

Figure 5BC. The expression of matrix metalloproteinases (MMPs) in Wharton’s jelly MSCs induced by various concentrations of AuNP after 48 hours. *p < 0.05, **p < 0.01, ***p < 0.001: compared to AuNP 0 ppm group.

Figure 6B. Wharton’s jelly MSCs migration ability with AuNP treatment for 48 hours. *p < 0.05, **p < 0.01, ***p < 0.001: compared to AuNP 0 ppm group.

Figure 7B-F. Expression of cell apoptotic-related proteins in Wharton’s jelly MSCs with various treatments for 48 hours. *p < 0.05, **p < 0.01, ***p < 0.001: compared to AuNP 0 ppm group.

Figure 8. The AuNP-Col uptake efficiency in Wharton’s jelly MSCs at 30 minutes, 2 hours, and 24 hours. *p < 0.05: compared to the AuNP 1.25 ppm group in IF method. *p < 0.05, **p < 0.01, ***p < 0.001: compared to the Control group in FACS method (treatment without AuNP-Col).

Figure 9. The endocytosis mechanisms of MSC uptake AuNP-Col (2.5 ppm) investigated by various inhibitors. *p < 0.05, **p < 0.01, ***p < 0.001: compared to the Control (treatment without inhibitors).

Figure 10. The intracellular transportation of AuNP-Col in Wharton’s jelly MSCs investigated by LysoTracker assay at 30 minutes, 2 hours, and 24 hours. **p < 0.01, ***p < 0.001: compared to the 30 min group.

Figure 11. The foreign body response induced by AuNP 2.5 ppm by in vivo animal models. **p < 0.01: compared to the Control (treatment without AuNP 2.5 ppm).

  1. How to explain that the viability of MSCs was enhanced by pure AuNP 1.25 ppm and AuNP 2.5 ppm? Why does the expression intensity decrease as the concentration of Au NPs increases? What are the reasons for thesephenomena?

Answer:
Thanks for the valuable comment of reviewer. We have addressed the important issue in the “Discussion” section. (Page 19, Line 654-660) The SEM images for different concentration of AuNP are displayed below.

“The effect of AuNP on MSC had better proliferation was lower for those containing either a lesser (1.25 and 2.5 ppm) amount of AuNP, suggesting that surface morphology may be more relevant homogenous distribution which may account for better cellular adhesion, migration effect [40]. However, the contribution of overloading of AuNP as a result of aggregation. The effect at higher AuNP (5 and 10 ppm) was not evident, possibly due to the aggregation of the nanoparticles. If the dispersion of AuNP could be improved, the effect could be even more pronounced at higher AuNP concentrations.”

Reference:

(40) Lin, R.-H.; Lee, H.-T.; Yeh, C.-A.; Yang, Y.-C.; Shen, C.-C.; Chang, K.-B.; Liu, B.-S.; Hsieh, H.-H.; Wang, H.-M.D.; Hung, H.-S. Favorable Biological Performance Regarding the Interaction between Gold Nanoparticles and Mesenchymal Stem Cells. International Journal of Molecular Sciences 2022, 24, 5.

  1. The discussion of the results is poor. In fact, the Discussion section contains a summary of the results. Besides, any comparison of the obtained data with the literature is lack. The discussion section should be improved. An improvement of the discussion is needed also in relation to similar or analogous results from literature and the explanation of possible differences.

Answer:
Thanks for the kindly suggestion from the Reviewer. We have included the more detail description in the “Discussion” section.

(1) “Current state of nanotechnology including metallic particles, liposomes, micelles, hydrogels and polymeric nanoparticles are applied for biomedical approaches [51]. However, the drawbacks such as cytotoxicity, induction of inflammatory responses, limited targeting efficiency and expensive for development need to be overcome [51]. The titanium (Ti) nanoparticles have been verified to induce inflammations and DNA damage leading to cell apoptosis in various cell lines [52, 53]. Moreover, the titanium dioxide (TiO2) nanoparticles were reported to cause injury in organs such as brain and liver due to oxidative DNA damage [54, 55]. Meanwhile, the porous silica nanoparticles have been studied for medical approaches, but the silanol groups on the surface of porous silica nanoparticles could interact with the phospholipids of erythrocyte to cause hemolysis [56]. For the polymeric nanoparticles such as PLGA-based or PCL-based delivery system, they have been reported to be hydrophobicity and need longer degradation time [57].” (Page 20, Line 708-719)

(2) “Additionally, our literatures have verified that AuNP modified with collagen is a biocompatible nano delivery system to carry natural compounds and demonstrates good anti-cancer capacities [62, 63]. However, the higher concentration of AuNP may lead to cytotoxicity, the mission to select appropriate concentration of nanoparticles to develop nanodrug delivery system becomes important for regenerative therapies [40].” (Page 21, Line 727-732)

(3) “Stem cells are widely applied for clinical therapies such as neurodegenerative diseases, including neural stem cells (NSCs), embryonic stem cells (ESCs) and adipose-derived stem cells (ASCs) [64]. However, several disadvantages such as difficult to obtain, potential of tumor formation and lack of self-renewal ability [65, 66]. Wharton’s jelly MSCs have several advantages to be the candidate for stem cell therapy owing to the higher expression of Oct4 and Sox2 pluripotency markers and anti-inflammatory abilities [67].” (Page 21, Line 733-738)
